# ACEIC: a comprehensive anthropogenic chlorine emission inventory for China

Siting Li[1,2], Yiming Liu[1,2,*], Yuqi Zhu[1,2], Yinbao Jin[1,2], Yingying Hong[3], Ao Shen[1,2], Yifei Xu[1,2], Haofan Wang[1,2], Haichao Wang[1,2], Xiao Lu[1,2], Shaojia Fan[1,2], Qi Fan[1,2,*]

[1]School of Atmospheric Sciences, Sun Yat-sen University, and Key Laboratory of Tropical Atmosphere-Ocean System, Ministry of Education, Zhuhai, China

[2]Guangdong Provincial Observation and Research Station for Climate Environment and Air Quality Change in the Pearl River Estuary, Southern Marine Science and Engineering Guangdong Laboratory (Zhuhai), Zhuhai, China

[3]Guangdong Ecological Meteorology Center, Guangzhou, China

*Correspondence to*: Yiming Liu (liuym88@mail.sysu.edu.cn), Qi Fan (eesfq@mail.sysu.edu.cn)

**Abstract.** Chlorine species play a crucial role as precursors to Cl radicals, which can significantly impact the atmospheric oxidation capacity and influence the levels of trace gases related to climate and air quality. Several studies have established the chlorine emission inventory in China in recent years, but the emission remains uncertain and requires further investigation. Anthropogenic Chlorine Emission Inventory for China (ACEIC) was the first chlorine emission inventory for China based on local data developed in our previous study, which only includes the emissions from coal combustion and waste incineration. In this study, we updated this inventory to include data for a more recent year (2019) and expanded the range of species considered (HCl, fine particulate $Cl^-$, $Cl_2$, and hypochlorous acid (HOCl)) as well as the number of anthropogenic sources (41 specific sources). Compared with previous studies, this updated inventory considered more anthropogenic sources, used more localized emission factors, and adopted more refined estimation methods. The total emissions of HCl, fine particulate $Cl^-$, $Cl_2$, and HOCl in mainland China for the year 2019 were estimated to be 361 (-18%~27%), 174 (-27%~59%), 18 (-10%~15%), and 79 (-12%~18%) Gg, respectively. To facilitate analysis, we aggregated the chlorine emissions from various sources into five economic sectors: power, industry, residential, agriculture, and biomass burning. HCl emissions were primarily derived from industry (43%), biomass burning (38%), and residential (13%) sectors. The biomass burning and industry sectors accounted for 74% and 19% of the fine particulate $Cl^-$ emissions, respectively. Residential and industry sectors contributed 61% and 29% of the total $Cl_2$ emissions. HOCl emissions were predominantly from the residential sector, constituting 90% of the total emissions. Notably, the usage of chlorine-containing disinfectants was identified as the most significant source of $Cl_2$ and HOCl emissions in the residential sector. Geographically, regions with high HCl and fine particulate $Cl^-$ emissions were found in North China Plain, Northeast China, Central China, and Yangtze River Delta, whereas the Pearl River Delta, Yangtze River Delta, and Beijing-Tianjin-Hebei regions exhibited elevated levels of $Cl_2$ and HOCl emissions. Regarding monthly variation, emissions of HCl and fine particulate $Cl^-$ were relatively higher during early spring (February to April) and winter

(December to January) due to intensified agricultural activities, while $Cl_2$ and $HOCl$ emissions were higher in the summer months due to increased demand for water disinfection. We incorporated this emission inventory to the chemical transport model and found the simulated concentrations of chlorine species agreed reasonably with the observations, which suggested the relatively faithful estimations of their emissions. This updated inventory contributes to a better understanding of anthropogenic sources of chlorine species and can aid in the formulation of emission control strategies to mitigate secondary pollution in China.

# 1 Introduction

Recent field and laboratory studies have revealed the crucial role of chlorine radical (Cl) in tropospheric chemistry (Faxon and Allen, 2013; Qiu et al., 2019a; Young et al., 2014; Peng et al., 2022). As highly reactive radicals, Cl can significantly impact the abundance of trace gases related to climate and air quality. Specifically, in the lower troposphere, Cl can initiate the oxidation of volatile organic compounds (VOCs), elevate the levels of conventional radicals (OH, $HO_2$, and $RO_2$), and produce ozone ($O_3$) and secondary aerosols that contribute to air pollution and alter the earth's radiation budget and climate (Qiu et al., 2019a; Wang et al., 2019; Li et al., 2021; Wang et al., 2020b). Furthermore, Cl reacts rapidly with methane, which is the most abundant hydrocarbon and the second-most important greenhouse gas emitted into the atmosphere (Li et al., 2022; Strode et al., 2020). Consequently, the study of chlorine chemistry in the troposphere has garnered increasing attention within the atmospheric chemistry community.

Chlorine species from anthropogenic activities, including HCl, fine particulate $Cl^-$ (pCl), $Cl_2$, and HOCl, are important precursors of Cl radicals. HCl can react with OH radicals to release Cl radicals (Riedel et al., 2012). Particulate $Cl^-$ provides aerosol surfaces for heterogeneous reactions with $N_2O_5$, producing $ClNO_2$ that is rapidly photolyzed into Cl radicals after sunrise (Thornton et al., 2010; Bertram and Thornton, 2009; Roberts et al., 2009; Osthoff et al., 2008). During the daytime, $Cl_2$ can be swiftly photolyzed, releasing two Cl radicals, while HOCl can also be photolyzed to release Cl radicals (Finlayson-Pitts, 1993; Faxon and Allen, 2013). These chlorine species are emitted from various anthropogenic sources (Chang and Allen, 2006; Yin et al., 2022), posing challenges in estimating their emissions. The development of chlorine emission inventories would enhance our understanding of the emission characteristics and primary sources of these emissions. Furthermore, it would provide crucial input data for numerical simulations of air quality, ultimately improving the accuracy of atmospheric pollutant predictions.

The study of estimating the anthropogenic chlorine emission in China began by Mcculloch et al. (1999), who established a global anthropogenic chlorine emission inventory called Reactive Chlorine Emissions Inventory (RCEI) based on a relatively rough statistical dataset in 1990. However, due to the rapid industrial and economic growth, this inventory cannot accurately represent the current situation of atmospheric chlorine emissions in China. The first anthropogenic chlorine emission inventory using local data in China (Anthropogenic Chlorine Emission Inventory for China (ACEIC)) was developed by our team (Liu et al., 2018), which considered the HCl and $Cl_2$ emissions from coal combustion and waste incineration for the year 2012. We then updated this inventory to the year 2014 (Hong et al., 2020). A more detailed chlorine emission inventory for the same year was established by Fu et al. (2018), including the HCl and pCl emissions from coal combustion, industrial production

processes, biomass combustion, and waste incineration. Zhang et al. (2022) also developed a comprehensive global emission inventory of HCl and pCl from 1960 to 2014, including China. Qiu et al. (2019b) compiled an updated emission inventory for Beijing in 2014 with pCl emissions from cooking considered. Lately, a team from Shanghai University has successively established anthropogenic chlorine emission inventories for Shanghai (Yi et al., 2020; Li et al., 2020b), Yangtze River Delta region (Yi et al., 2021), and China (Yin et al., 2022), with the consideration of more chlorine species (HCl, pCl, $Cl_2$, and HOCl) from various sources.

Despite these studies, the anthropogenic chlorine emission in China remains uncertain and further investigation is still warranted. Firstly, the estimated chlorine emissions in China varied in different studies due to the different applications of emission factors and estimation methods. Some studies have utilized emission factors derived from foreign sources or standards and guidelines that may not accurately reflect the specific local conditions in China. Some calculation methods are rudimentary and lack the granularity needed to effectively capture variations among provinces or different sources. Secondly, some modeling studies (Choi et al., 2020; Li et al., 2021) have used the anthropogenic chlorine emission as inputs and found that the simulated concentrations of chlorine species (HCl and pCl) were underestimated against the observation, which suggests that there are large uncertainties or missing sources for the current emission estimation. Lastly, chlorine emissions from anthropogenic activities were reported in the recent literature, but they have not been considered in the developed emission inventory for China. Such overlooked emissions include those from medical waste incineration, environmental disinfection, tap water utilization, pesticide application, and etc. The neglect of these sources can lead to an underestimation of the total chlorine emission. As a result, the development of a comprehensive anthropogenic chlorine emission inventory that addresses the above issues is of great significance in reducing the uncertainty of the emission estimation.

In this study, we update and improve the ACEIC inventory we developed previously to a more comprehensive one, which includes 4 chlorine species from 41 specific anthropogenic sources, with 2019 as the base year. Compared with the previous studies, this updated inventory considered more anthropogenic sources, used more localized emission factors, and adopted more refined estimation methods, which aims to reduce the uncertainty of chlorine emission estimations. We also incorporated the emission inventory to a chemical transport model to evaluate the accuracy of emission estimations. The structure of this paper is organized as follows. Section 2 demonstrates the data and method utilized to develop the updated chlorine emission inventory ACEIC. Section 3 presents the characteristic of anthropogenic chlorine emissions, and Section 4 provides the validation, uncertainty and limitation of this inventory. The conclusion is summarized in Section 5.

**2 Data and Method**

**2.1 Emission estimation**

In our study, we have compiled the emissions of chlorine species (HCl, pCl, $Cl_2$, HOCl) from 41 anthropogenic activities (Table S1 and S2) across the 31 provinces in mainland China. These emissions are categorized into seven major source categories (Table S1): (1) coal combustion, (2) industrial production processes, (3) waste incineration, (4) biomass burning, (5) cooking, (6) usage of chlorine-containing disinfectants, and (7) usage of pesticides. To estimate anthropogenic chlorine emissions in China for the year 2019, we have employed the "emission factor" method, which is represented by the following equation.

$$E_{HCl} = \sum_{i,j} A_{i,j} \times EF_{(HCl)i,j} \tag{1}$$

$$E_{pCl} = \sum_{i,j} A_{i,j} \times EF_{(PM_{2.5})i,j} \times M_{i,j} \tag{2}$$

$$E_{Cl_2/HOCl} = \sum_{i,j,k} A_{i,j} \times \left(CD_{i,j} - CR_{i,j}\right) \times f_j \times R_k \tag{3}$$

Generally, the estimation of HCl and pCl emissions was carried out using formulas 1 and 2, respectively, while the calculation of $Cl_2$ and HOCl emissions utilized formula 3. In these formulas, the variables i, j, and k represent provinces, emission sources, and chlorine precursors, respectively. The symbol E represents emissions, A represents activity data, and EF represents emission factors. The variable M in equation 2 represents the proportion of $Cl^-$ in $PM_{2.5}$. For equation 3, the variable CD represents the amount of chlorine added, CR represents the residual chlorine, f represents the volatilization rate of chlorine, and R represents the release ratios of chlorine gases.

**2.2 Activity data and emission factor**

The activity data and emission factor used to calculate the chlorine emissions of each source category are demonstrated in this section. This emission inventory uses a large amount of activity data and advanced emission factors. Most of the activity data, emission factors, and related references can be found in Tables S3-S15. Generally, the activity data were obtained from the yearbook (e.g., China Energy Statistical Yearbook, China Industry Statistical Yearbook, and China Urban-Rural Construction Statistical Yearbook), government statistics (e.g., National Bureau of Statistics, and General Administration of Sport of China), and Gaode's POI data. The emission factors were mainly based on the measured and survey data from the literature. The selection process of emission factors was guided by the principles of prioritizing domestic local areas, prioritizing the most

recent year, and giving precedence to field observations. By adhering to these principles, the study seeks to minimize the inherent uncertainty associated with emission factors to the greatest extent possible.

**2.2.1 Coal combustion**

Chlorine in the coal can be released into the atmosphere through coal combustion. We estimated the emissions from coal combustion based on the coal consumption data, which were derived from the provincial energy balance spreadsheets of the "China Energy Statistical Yearbook" (National Bureau of Statistics, 2020c). We classified coal consumption into 4 economic sectors: (1) power plants. (2) industry, including industrial processes, construction processes, and heating plants. (3) residential, including residential and commercial activities, transportation, and others. (4) agriculture. The emissions from coal combustion were calculated using the formula proposed in our previous study (Liu et al., 2018):

$$E_{i,j,k} = A_{i,j} \times c_i \times \sum_k \left[ X_{j,k} \times R_{j,k} \times \left(1 - \eta_{d_{j,k}}\right) \times \left(1 - \eta_{s_{j,k}}\right) \right] \times \rho \times \frac{1}{MM_k} \tag{4}$$

where i, j, k, l are the province, source sector, chlorine species, and the energy allocation type (type of boiler and control device combination). A is the coal consumption by province in each source category, and c is the chlorine content in coal by province. Instead of using chlorine content in raw coal produced in each province (Liu et al., 2018), we adopted the data from the study of Fu et al. (2018), which is the value of consumed coal considering the coal transportation. X is the fraction of energy for a sector (energy allocation ratio), and R is the chlorine release rate. $\eta_d$ is the removal efficiency of dust-removal facilities, and $\eta_s$ is the removal efficiency of sulfate-removal facilities. The values of X, R, $\eta_d$, and $\eta_s$ can be found in Table 2 of our previous study (Liu et al., 2018). $\rho$ is the chlorine proportion of HCl (86.33%), fine particulate Cl$^-$ (10.09%), and Cl$_2$ (3.58%) in emitted flue gases based on the local measurement (Deng et al., 2014). MM refers to mass ratio of chlorine in chlorine species (35.5/36.5, 1, 1 for HCl, pCl, and Cl$_2$).

**2.2.2 Industrial production process**

Chlorine species can be released during some industrial production processes. For example, the production of cement, iron, and steel will emit HCl and fine particulate Cl$^-$, and HCl will be volatilized during the production of hydrochloric acid, while the production of flat glass will produce HCl, fine particulate Cl$^-$, and Cl$_2$. The emissions from these industrial production processes were estimated based on their production. The data of cement, iron, steel, and flat glass by province were collected from China Industry Statistical Yearbook (National Bureau of Statistics, 2020e), and those of industrial HCl were obtained from the National Bureau of Statistics (https://m.sohu.com/a/335035620_775892/?pvid=000115_3w_a, last access: 1 August 2024).

For HCl and $Cl_2$, the emissions were calculated using formula 1. The HCl emission factors of industrial productions of cement, iron, steel, hydrochloric acid, and flat glass were obtained from the literature (see Table S4). The $Cl_2$ emission factor of flat glass is 0.58 g $t^{-1}$ (Sepa, 2011; Wang et al., 2014; Zheng et al., 2018). For fine particulate $Cl^-$, the emission factors ($EF_{pCl}$) were calculated from the emission factor of $PM_{2.5}$ ($EF_{PM2.5}$) and the $Cl^-$ proportion in $PM_{2.5}$ (M):

$$EF_{pCl,i} = EF_{PM_{2.5},i} \times M_i \tag{5}$$

where i denotes different industrial production processes. These parameters for the industrial production of cement, iron, steel, and flat glass are shown in Table S5.

### 2.2.3 Waste incineration

The primary waste disposal method in China is the waste incineration, which releases HCl and pCl during the incineration process. While China prohibits the open burning of MSW (municipal solid waste), there are still instances of open incineration in certain areas. Apart from MSW waste, the incineration of medical waste can emit chlorine-containing species due to its high plastic content. Therefore, waste incineration can be categorized into three types: MSW incineration stations, MSW open incineration, and medical waste incineration.

For the emission from incineration stations, the waste for incinerations in each province/city were collected from the China Urban-Rural Construction Statistical Yearbook (National Bureau of Statistics, 2020d). The emissions of HCl and pCl are calculated using formulas 1 and 2. We estimated the HCl and $PM_{2.5}$ emission factors to be 0.0259 and 0.0136g $kg^{-1}$, respectively, for 2019 based on the proportion of incinerator types used, the application of air pollution control devices, and their emission factors in the study of Fu et al. (2022), as shown in Tables S4 and S5.

For the MSW open incineration, there is no relevant statistical data at present, so its emission was estimated as follows.

$$A_i = \left(P_{i,u} \times W_u \times F_{i,u} + P_{i,r} \times W_r \times F_{i,r}\right) \times B \times 365 \tag{6}$$

where i, u, and r represent the province, urban, and rural data, respectively. P is the population. W is the per capita waste production rate, which is 1.2kg $person^{-1}$ $d^{-1}$ in urban areas and 0.79 kg $person^{-1}$ $d^{-1}$ in rural areas (Wang et al., 2017b; He et al., 2010). F represents the proportion of open burning of solid waste, which means the untreated portion (1-f). f represents the

treated proportion of solid waste, which is derived from the China Urban and Rural Construction Statistical Yearbook 2019 (National Bureau of Statistics, 2020d). The F value varied in different provinces due to the imbalance of economy, urbanization, and waste disposal technology popularization. B is the waste combustible rate, which is assumed to be 0.6 based on the study conducted by Fu et al. (2018). The emission factors of HCl and fine particulate Cl⁻ are shown in Table S4 and S5.

For the medical waste incineration, the national production of medical waste can be obtained from Ruiguan.com (https://www.reportrc.com/article/20200506/6615.html, last access: 1 August 2024). It is reported that the inpatient and outpatient departments produced 1.92 and 340000 tons of medical waste, respectively. The production of medical waste in inpatient and outpatient departments in various provinces can be estimated based on formulas 7 and 8, respectively:

$$IW_i = \frac{B_i \times U_i}{B_{sum} \times U_{sum}} \times IW_{sum} \tag{7}$$

$$OW_i = \frac{P_i}{P_{sum}} \times OW_{sum} \tag{8}$$

where i and sum represent the province and national total, respectively, while IW and OW represent the production of medical waste in the inpatient and outpatient departments, respectively. B represents the number of beds, U represents the utilization rate of beds, and P represents the number of visits, all of which can be obtained from the China Health Statistics Yearbook 2020 (National Health Commission of the People's Republic of China, 2020).

We estimate the HCl emission factor of medical waste based on its component proportion and HCl release rate, using the following formula:

$$EF_i = \sum (N_i \times R_i) \times (1 - \eta_d) \times (1 - \eta_s) \tag{9}$$

where i represents different components, N represents the proportion of components, and R represents the HCl release rate, as shown in Table S6. $\eta_d$ is the chlorine removal efficiency of dust-removal facilities (25.1%), and $\eta_s$ is the chlorine removal efficiency of sulfate-removal facilities (95.5%) (Liu et al., 2018). Due to the lack of relevant data on pCl emission factors for medical waste incineration, we set it to $EF_{HCl}$ (medical)/12.8 based on the $EF_{HCl}/EF_{pCl}$ for the MSW incineration stations being 12.8.

**2.2.4 Biomass burning**

Biomass burning includes household burning and open burning. Both of them include straw burning and firewood burning. The emissions from straw burning were calculated as follows:

$$A_{i,j,k} = P_{i,j} \times R_j \times D_j \times C_j \times F_{i,j,k} \tag{10}$$

where i, j, and k represent respectively the province, crop type, and combustion type. P represents the crop yield, R signifies the ratio of straw to crop products, and D represents the dry matter ratio. C denotes the combustion efficiency, and F represents the rates of household and open combustion. This study recalibrated the open burning ratio based on the variation in fire radiative power over croplands from MODIS satellite fire point data (https://modis.gsfc.nasa.gov, last access: 1 August 2024), re-adjusting the ratios initially proposed in the study of Zhou et al. (2017). Additionally, utilizing the research by Liu et al. (2022), we estimated the household burning ratio for the year 2019 through statistical linear trend analysis. Specific values for these parameters can be found in Tables S7 and S8.

For the firewood burning, the emissions were estimated using the Eq. 11:

$$A_i = \frac{P_i}{S_i} \times B \times T \tag{11}$$

where i represents the province. P represents the rural population, while S represents the average size of rural households. The division P/S represents the number of rural households. B represents the consumption rate of firewood, and T represents the number of days on which firewood is burned. According to the research of Yi et al. (2021), it is assumed that each rural household burns 2 kg of firewood per day, and the total number of burning days in a year is 260. The emission factors of HCl and fine particulate Cl⁻ from straw and firewood combustion are shown in Tables S4 and S9, respectively.

**2.2.5 Cooking**

Chlorine emissions from cooking are mainly due to the usage of edible salt, which is released in the form of aerosols during the cooking process (Zhang et al., 2017). The emissions from cooking activities were categorized into three types: household catering, social catering, and canteen catering. In social catering, small and medium-sized catering enterprises contribute to more than 80% of the total emissions (Wu et al., 2018). Canteen catering includes school canteens and unit canteens. The formula for estimating pCl emissions from cooking was as follows:

$$E_{i,j} = N_{i,j} \times n_j \times Q_j \times H_j \times D_j \times (1 - \eta) \times EF_{PM2.5,j} \times M \tag{12}$$

where i and j represent provinces and catering types, respectively. N represents the number of households (i.e. population/family size), the number of social restaurants, the number of students and faculty members in middle schools and colleges, and the number of public institutions and government agencies. n is the number of furnaces, and Q is the smoke emission. H represents the cooking time per day, D is the cooking days, and η is the removal efficiency of the flue gas scrubber.

$EF_{PM2.5}$ is the emission factor of $PM_{2.5}$, the above parameters are based on the study of Wu et al. (2018). M is the proportion of $Cl^-$ in $PM_{2.5}$. According to the local measurement of Li et al. (2018), the average proportion of $Cl^-$ in $PM_{2.5}$ is 1.545%. The parameters related to the emission factors of cooking are shown in Table S10.

**2.2.6 Usage of chlorine-containing disinfectant**

$Cl_2$ and HOCl released from the usages of chlorine-containing disinfectant are known for their pungent odors, which can pose risks to human health. Consequently, indoor disinfection generally requires meticulous attention to ventilation, such as in hospitals and indoor swimming pools (Huang, 2012; Tang, 2003; the standard GB 15982-2012). The rapid air exchange between indoor and outdoor during the ventilation process can help the emissions of chlorine gases into the atmosphere.

The emissions from the disinfection process were estimated as follows:

$$E_{i,j} = W_i \times (CD - CR) \times f \times R_j \tag{13}$$

where i and j are the province and chlorine species, respectively. W is the amount of water for disinfection treatment. CD and

CR are the chlorine dose added to the water and the residual chlorine after disinfection, respectively. f is the volatilization ratio of chlorine, and R is the release ratio of chlorine gases. In this study, the release ratios of HOCl and $Cl_2$ during the disinfection process are 84% and 11%, respectively (Wong et al., 2017). Except for the activity data (W), other parameters in this equation for different usages of disinfectant are presented in Table S11.

a Cooling tower

To prevent the breeding of bacteria and algae in the circulating water of the cooling tower and reduce the cooling efficiency, it is necessary to add disinfectant regularly and maintain a certain concentration. The volatilization of cooling water will cause

the release of chlorine gases into the atmosphere. The chlorine emission was estimated based on the amount of supplementary water that was regularly added to the cooling tower. It was estimated according to the following equation:

$$W_i = A_i \times L \times S \tag{14}$$

where i is the province, and A is the industrial water consumption. These provincial activity data were obtained from the China Environmental Statistics Yearbook 2020 (National Bureau of Statistics, 2020a). L is the proportion of cooling water, which accounts for about 60% of industrial water (Hou and Zhang, 2015). S is the proportion of supplementary water. Here we used the value of 2.4% (Wang et al., 2020a; Zhao, 2015).

b Water treatment

During the water treatment process, which typically involves coagulation, sedimentation, filtration, and disinfection, the use of chlorine-containing disinfectants is a common practice for effective disinfection (Ge et al., 2006). As a result of this disinfection process, emissions of HOCl and $Cl_2$ are released into the atmosphere. The emission of water treatment is estimated using formula 13. The water amount that needs disinfection ($W_i$) is the quantity of tap water supplied in each province, which can be obtained from the China Urban and Rural Construction Statistical Yearbook 2020 (National Bureau of Statistics, 2020d). The emission factors can be found in Table S11. The residual chlorine at the end of the pipeline network amount can be obtainted from the website of water bureau for each province, as shown in Table S12.

c Waste water treatment

Waste water treatment encompasses the treatment of both medical waste water and domestic sewage. Before discharge, medical waste water undergoes disinfection to prevent the dissemination of pathogens. Similarly, domestic sewage is disinfected before being released into natural water bodies to safeguard the ecological balance against the proliferation of algae and microorganisms. The treatment capacity for medical waste water can be estimated based on the number of hospital beds and the rate of waste water production.

$$W_i = N_i \times Q \times D \tag{15}$$

where i is the province, and N is the number of hospital beds, which was derived from the China Health Statistics Yearbook 2020 (National Health Commission of the People's Republic of China, 2020). Q is the waste water production rate, each bed produces 0.62 m³ of medical waste water per day (Zhou, 1987). D is the number of days of disinfection, which is 365.

The emissions from domestic waste water were estimated based on the provincial sewage treatment capacity. These data were obtained from the China Urban and Rural Construction Statistical Yearbook 2020 (National Bureau of Statistics, 2020d).

d Swimming pool

To inhibit bacterial growth and ensure a sanitary environment in swimming pools, regular disinfection of the pool water is

315 necessary (Wang et al., 2002). Due to the lack of data on private swimming pools, this study only considers emissions from public swimming pools. The volume of public swimming pools was calculated as follows:

$$V_i = n_i \times \sum_j (a_j \times b_j \times h_j \times r_j) \tag{16}$$

where i and j represent different provinces and size types. Swimming pool size types include standard, semi-standard/non-standard swimming pools. We assume that the sizes of semi-standard and non-standard swimming pools are the same. n is the number of swimming pools, and the provincial data comes from the State Sports General Administration (https://www.sport.gov.cn/, last access: 1 August 2024). a, b, and h are the length, width, and depth of the swimming pool with different size types, as shown in Table S13. r represents the proportion of different size types of swimming pools, with standard

swimming pools accounting for 28%, and semi-standard/non-standard swimming pools accounting for 72% (Zhang, 2015).

The HOCl and $Cl_2$ emissions of the swimming pool are calculated:

$$E_{i,k,l} = V_i \times \sum_k (y \times z \times D) \times c \times f \times R_l \tag{17}$$

where i, k, and l are provinces, indoor and outdoor types (outdoor and indoor swimming pools), and chlorine species, respectively. y is the proportion of indoor and outdoor types, with outdoor swimming pools accounting for 59.43% and indoor swimming pools accounting for 38.93% according to the survey of State Sports General Administration

(https://www.sport.gov.cn/, last access: 1 August 2024). z for the dosage, according to the research of Wang et al. (2002) and

Qianzhan Industrial Research Institute (https://bg.qianzhan.com/report/detail/300/210803-5009dc2a.html, last access: 1 August 2024), the outdoor swimming pool uses 2.10g m$^{-3}$ of strong chlorine per day, and the indoor swimming pool uses 1.31g m$^{-3}$ every day. D is the number of opening days for the swimming pool. The indoor swimming pool in this study is open all year round, and the outdoor swimming pool is only open in summer. c is the mass fraction of available chlorine in strong chlorine, which is 90% (Wang et al., 2002). f is the chlorine volatilization ratio, which is 0.2 (Li et al., 2020b). R is the release ratio of HOCl (84%) and Cl$_2$ (11%) (Wong et al., 2017).

e Environmental disinfection

Environmental disinfection includes hospital disinfection, breeding disinfection, and toilet disinfection. For hospital disinfection, chlorine emissions are related to the number of hospitals and the amount of disinfectant used, and the emissions are as follows:

$$E_{i,k} = n_i \times U \times c \times f \times R_k \tag{18}$$

where i and k represent provinces and chlorine-containing species, and n is the number of hospitals. U denotes the quantity of disinfectant utilized. In 2007, the average quantity of chlorine-containing disinfectant used in five Taizhou hospitals was reported to be 2329.2L (Sun et al., 2007). Due to the absence of data in 2019, we assumed that its change is proportional to the total health cost in recent years. The amount of disinfectant usage is estimated using the formula $U_{2019}=U_{2007} \times C_{2019}/C_{2007}$, where U represents the amount of disinfectant usage, and C represents the total health cost. The total health cost can be obtained from the China Health Statistical Yearbook (National Health Commission of the People's Republic of China, 2020, 2008). As a result, the usage of disinfectants in 2019 is estimated to be 13250.2 L. c is the chlorine content of the disinfectant, which is 1000 mg/L. f is the chlorine volatilization ratio, which is 0.3 (Li et al., 2020b). R is the release ratio of HOCl (84%) and Cl$_2$ (11%) (Wong et al., 2017).

Breeding disinfection is found in pig farming, poultry farming, and aquaculture. The chlorine emissions from these sources are calculated using the following formula:

$$E_{i,j,k} = S_{i,j} \times U_j \times N_j \times r \times f \times R_k \tag{19}$$

where i, j, and k represent provinces, different breeding types, and chlorine-containing species. S is the breeding area. The breeding areas of pigs and poultry are calculated by the number of pigs and poultry and the breeding density. The number of pigs and poultry and the aquaculture area can be obtained from statistical data (National Bureau of Statistics, 2020b). U is the amount of disinfectant used per unit area, and N is the disinfection frequency. f is the chlorine volatilization ratio, and these parameters can be found in Table S14. r is the proportion of chlorine-containing disinfectants used for breeding, which is 0.065 (Jing et al., 2019). R is the release ratio of HOCl (84%) and $Cl_2$ (11%) (Wong et al., 2017).

Toilet disinfection includes disinfection of public toilets and household toilets. Estimates of emissions during the disinfection of public toilets are as follows:

$$E_{i,k} = n_i \times U \times c \times N \times f \times R_k \tag{20}$$

where i and k represent provinces and chlorine-containing species, and n is the number of public toilets. U is the daily disinfectant dose of each toilet, which is 523.8 mL (Wong et al., 2017) (https://bg.qianzhan.com/report/detail/300/210803-5009dc2a.html, last access: 1 August 2024). c is the disinfectant concentration, which is 5%. N is the disinfection frequency, which is 365 times/year. f is the chlorine emission rate, which is 0.3. The parameters are all based on the study by Li et al. (2020b). R is the release ratio of HOCl (84%) and $Cl_2$ (11%) (Wong et al., 2017). There is too little relevant data on household toilet disinfection, and its emission is estimated to be twice that of public toilets (Li et al., 2020b).

f Tap water use

After water treatment, there is still residual chlorine in the tap water at the end of the pipe network, the residual chlorine will be released into the atmosphere during the use of tap water. In this study, we considered the emissions from car washing, lawn watering, road sprinkling, and pipe leaks.

For the car washing, the chlorine emissions were estimated using the following formula:

$$E_{i,j} = \sum_k (S_{i,k} \times U_k \times N_k) \times CR \times R_j \tag{21}$$

where i, j, and k represent the province, chlorine species, and car types, respectively. S is the number of cars, which can be obtained from the National Bureau of Statistics (https://www.stats.gov.cn/, last access: 1 August 2024). U is the water

consumption per vehicle, N is the frequency of water use, and these parameters can be found in Table S15. Except for medium and large passenger cars, we assume that other cars are washed once a week. CR is the amount of residual chlorine, which can be found in Table S12, and R is the release ratio of HOCl (84%) and $Cl_2$ (11%) (Wong et al., 2017).

For the lawn watering, road sprinkling, the chlorine emissions were estimated using the following formula:

$$E_{i,j,k} = S_{i,j} \times U_j \times N_j \times CR \times R_k \tag{21}$$

where i, j, and k represent the province, source category, and chlorine species, respectively. S is the number of green area, and road area. U is the unit water consumption, the unit water consumption of lawn watering is $1.5L/m^2$, and road sprinkling is $1L/m^2$ (Li et al., 2020b). N is the frequency of water use. Lawn watering is 45 times, and road sprinkling is 50 times (Li et al., 2020b). CR is the amount of residual chlorine, which can be found in Table S12, and R is the release ratio of HOCl (84%) and $Cl_2$ (11%) (Wong et al., 2017).

For the pipeline leaks, the chlorine emissions are estimated as follows:

$$E_{i,k} = W_i \times CR \times f \times R_k \tag{22}$$

where i and k represent provinces and chlorine-containing species. W is the amount of water loss. CR is the amount of residual chlorine, which is 0.86 mg/L (Li et al., 2020b). f is the chlorine volatilization ratio, which is 0.1 (Li et al., 2020b). R is the release ratio of HOCl (84%) and $Cl_2$ (11%) (Wong et al., 2017).

### 2.2.7 Usage of pesticide

Active chlorine is also released during pesticide application, thereby enhancing the activity of the atmosphere. The pesticides in this study include insecticides and herbicides. The chlorine emissions from them are calculated using the following formula:

$$E_{i,j} = P_i \times O \times M \div \rho \times A \times c \times f \times R_j \tag{23}$$

where i and j represent provinces and chlorine-containing species, respectively. P is the amount of pesticide usage, O is the proportion of organochlorine pesticides, and the use of organochlorine pesticides accounts for 30% of pesticides (Zhang, 2016). M is the proportion of insecticides and herbicides, 68% and 23% respectively (Zhang, 2016). $\rho$ is the density, the density of

the insecticide is 1.359g/ml, the herbicide is mainly liquid, and the density value is 1. A is the active ingredient content, the insecticide is 94.8g/L, and the herbicide c is the chlorine content, the chlorine content of pesticides is 13.08%, and the chlorine content of herbicides is 11.48%. f is the chlorine volatilization ratio, which is 0.3. The above parameters are all based on the research of Yi et al. (2021). R is the release ratio of HOCl (84%) and $Cl_2$ (11%) during the pesticide application (Wong et al., 2017; Yi et al., 2021).

## 2.3 Spatial allocation

The estimated provincial chlorine emissions were further gridded into a $0.1° \times 0.1°$ grid to derive a gridded, model-ready emission inventory. The spatial allocation of emissions was handled separately according to point sources and area sources. For point sources with clear location identification, including coal-fired emissions from power plants, coal-fired emissions from heating, emissions from waste incineration stations, and biomass open combustion emissions, the point source emissions will be directly located in the grid according to the longitude and latitude coordinates of the point source. For the emissions from power plants and MSW incineration stations, the emissions in each province were distributed to each location according to its installed capacity. For the emission of biomass open burning, we allocated the provincial emissions spatially to the fire location according to its fire radiation power over the cropland. The fire location and its fire radiation power data were derived from the MODIS satellite data (https://modis.gsfc.nasa.gov, last access: 1 August 2024). For emissions from other point sources with unavailable installed capacity data, such as industrial production, we assumed a uniform emission for them and spatially allocated the provincial emissions evenly to each point. For the area sources, the provincial emissions were spatially disaggregated onto grid cells using empirically selected spatial proxies such as population density (total, urban, and rural). The detailed spatial allocation factors of each source category and their sources are listed in Table S16.

## 2.4 Temporal allocation

The annual chlorine emissions estimated above were further allocated to each month using the corresponding activity data or the selected monthly proxies. For the emissions of coal-fired power plants and cooling towers, the monthly emissions are allocated according to the thermal power generation. For the emission of heating coal, the time distribution coefficient is set from mid-November to mid-March according to the heating conditions in different regions of China. For the emissions from the industrial production process, pesticide use, and restaurants, the temporal allocation is based on the output of industrial products, pesticides, and revenue of the catering industry, respectively. The monthly output of thermal power generation,

industrial products, pesticides and revenue of the catering industry can be found in the National Bureau of Statistics (https://www.stats.gov.cn/, last access: 1 August 2024). Based on the fire location and its fire radiation power over the cropland from the MODIS satellite data, we performed temporal allocation of chlorine emissions from biomass burning for each province. According to the proportion of the number of outdoor and indoor swimming pools, it is assumed that 60% of the swimming pools are open from mid-May to mid-September, 40% of the swimming pools are open all year round, and the time distribution coefficient of the disinfection discharge of the swimming pool is set accordingly. For other coal burning and environmental disinfection emissions that are not sensitive to time change, the monthly emissions can be distributed by days. Other emissions will be allocated according to the existing research. The monthly allocation factors for each source category are presented in Table S17.

## 3 Results

### 3.1 Anthropogenic chlorine emission for China (ACEIC)

The ACEIC (Anthropogenic Chlorine Emission Inventory for China) inventory for the year 2019 was developed in this study. The general information of this inventory is shown in Table 1. It includes emissions from 31 provinces in mainland China. We estimated the total emissions of HCl, fine particulate $Cl^-$, $Cl_2$, and HOCl in mainland China to be 361, 174, 18, and 79 Gg, respectively. The estimated emissions by source category are presented in Table 2. Figure 1 shows the contribution of different source categories to the total emission. For HCl emission, coal combustion and biomass burning are the primary sources, accounting for 47% (170 Gg) and 38% (137 Gg) of the total, respectively. The emissions from industrial production processes and waste incineration make up 11% (39 Gg) and 4% (17 Gg), respectively. For fine particulate $Cl^-$, biomass burning is the major contributor to the total emission (74%, 128 Gg), followed by industrial production process (12%, 20 Gg), coal combustion (11%, 19 Gg), and waste incineration (3%, 5 Gg). For $Cl_2$, it is mainly from the usage of chlorine-containing disinfectants (59%, 10 Gg) and coal combustion (41%, 7 Gg). For HOCl emissions, the usage of chlorine-containing disinfectant is the major contributor (99%, 79 Gg). We note that the usage of chlorine-containing disinfectants is the major source of $Cl_2$ and HOCl. Figure 2 presents the proportion of different usages of chlorine-containing disinfectants. The waste water treatment account for 43% of the total, followed by water treatment (31%), environmental disinfection (13%), swimming pool (9%), tap water use (2%), and cooling tower (2%).

**3.2 Anthropogenic chlorine emission from different economic sectors**

We aggregated the anthropogenic chlorine emissions from 41 specific sources into 5 economic sectors, including power, industry, residential, agricultural, and biomass burning (Table S2). Table 3 provides the estimated emissions for HCl, fine particulate Cl⁻, $Cl_2$, and HOCl, while Figure 3 illustrates the contribution of each economic sector to the total emissions. HCl emissions were predominantly attributed to industry (43%, 155 Gg), biomass burning (38%, 137 Gg), and the residential sector (13%, 47 Gg). For fine particulate Cl⁻ emissions, biomass burning accounted for the majority with 74% (128 Gg), followed by the industry sector with 19% (33 Gg). $Cl_2$ emissions were primarily sourced from the residential sector (61%, 10 Gg) and the industry sector (29%, 5 Gg). As for HOCl emissions, the residential sector dominated, contributing 90% (72 Gg) of the total.

**3.3 Anthropogenic chlorine emission in different provinces**

Fig. 4 and Table 4 display the regional variations in anthropogenic chlorine emissions across different provinces. Regarding HCl emissions, Heilongjiang (36.76 Gg), Hebei (23.76 Gg), Shandong (22.12 Gg), Jilin (19.30 Gg), and Hunan (18.65 Gg) emerge as the top five contributing provinces. They account for 10.2%, 6.6%, 6.1%, 5.3%, and 5.2% of the total emissions, respectively. The elevated emissions in Heilongjiang and Jilin are attributed to the major contributions of biomass burning with higher agricultural production, which accounts for 83% and 84%, respectively. For Hebei, Shandong, and Hunan, the higher emissions from industrial production are the major contributors, accounting for 50%, 57% and 39%, respectively. The top five contributors to fine particulate Cl⁻ emissions are Heilongjiang (27.18 Gg), Jilin (14.29 Gg), Hebei (10.97 Gg), Jiangxi (10.53 Gg), and Henan (8.35 Gg). In these provinces, higher biomass burning emissions induced by active agricultural activities dominate the total emission. $Cl_2$ emissions are predominantly attributed to Guangdong (1.59 Gg), Shandong (1.31 Gg), Jiangsu (1.09 Gg), Hebei (1.02 Gg), and Hunan (0.96 Gg). The top five provinces contributing to HOCl emissions are Guangdong (9.92 Gg), Jiangsu (6.32 Gg), Shandong (5.34 Gg), Zhejiang (4.49 Gg), and Sichuan (3.87 Gg). Due to the large population and developed economy that stimulates the need for disinfection processes, provinces such as Guangdong, Shandong, and Jiangsu have relatively high emissions of $Cl_2$ and HOCl.

The spatial distribution of anthropogenic chlorine emissions reveals distinct patterns when considering per-unit-area (Fig. S1) and per-capita (Fig. S2) intensity by province. For the per-unit-area emission intensity, Shanghai has the highest emission intensity of HCl (171.68 kg km⁻²), $Cl_2$ (57.15 kg km⁻²), and HOCl (396.33 kg km⁻²), which is due to its small area. Jiangsu is the province with the highest emission intensity of fine particulate Cl⁻ (78.27 kg km⁻²), which is attributed to its relatively higher emission but smaller area. For the per-capita emission intensity, Heilongjiang has the highest emission intensity of HCl

(1129.46 g per people) and fine particulate Cl⁻ (834.90 g per people) due to its highest emission across the country. Ningxia is the province with the highest emission intensity of $Cl_2$ (37.68 g per people) due to its low population. Shanghai is the province with the highest emission intensity HOCl (101.28 g per people) due to its relatively higher emission but lower population.

## 3.4 Spatial distribution of the emission

Figure 5 shows the spatial distribution of anthropogenic chlorine emissions with a high-resolution granularity of 0.1° × 0.1°. Generally, eastern China exhibits significantly higher emissions compared to Western China. The distribution pattern of HCl and fine particulate Cl⁻ closely resemble one another, attributable to their primary sources: biomass burning and coal combustion. Similarly, $Cl_2$ and HOCl display comparable distribution patterns due to their predominant contribution from the usage of chlorine-containing disinfectants. Regarding HCl emissions, elevated levels are primarily concentrated in key regions

such as the North China Plain (NCP), the Northeast China, the Central China, and the Yangtze River Delta (YRD). The spatial distribution pattern of fine particulate Cl⁻ closely mirrors that of HCl emissions, except for slightly lower emissions observed in the PRD region. For $Cl_2$ and HOCl, regions with high emissions encompass the PRD, YRD, Beijing-Tianjin-Hebei region (BTH), SCB, and the central area of Hubei province. These regions exhibit higher population densities and economic development levels, which stimulate the demand for cleaning products and consequently contribute to elevated chlorine

emissions.

We also present the spatial distribution of anthropogenic chlorine in various energy sources (Fig. S3) and sectors (Fig. S4). In terms of energy consumption sources, coal emissions are mainly concentrated in North China, Central China, Southwest China, and Northwest China. The emissions from industrial production and waste treatment are mainly concentrated in the BTH,

YRD, PRD, and SCB. The spatial distribution of emissions from biomass combustion is similar to that of pCl emissions, while the emissions from disinfectant use are similar to the spatial distribution of $Cl_2$ and HOCl emissions. Concerning the sector-based distribution, the distribution of power plants is similar to the distribution of coal emissions. The emissions from industrial and civil sectors are mainly concentrated in BTH, YRD, PRD, and SCB. The agricultural sector is mainly concentrated in the southern region. Biomass combustion, as mentioned above, is similar to the distribution of pCl.

## 3.5 Temporal variation of the emission

Figure 6 shows the temporal variation of anthropogenic emissions for different chlorine species. For HCl and pCl, emissions are mainly concentrated in early spring (February to April) and winter (December to January). The high emission in these

months is attributed to the biomass burning emission with active agricultural activities. In contrast, emissions from other sectors remain relatively stable throughout the year. It's worth noting that the monthly variations vary across different regions because of the varied period of biomass burning, as shown in Fig. S5 and S6. For example, in Northeast China (Liaoning, Jilin, and Heilongjiang), where extensive straw burning occurs before crop planting, emissions are elevated in spring only.

The emissions of $Cl_2$ and HOCl show a different temporal variation pattern from those of HCl and pCl. Their monthly variations are primarily driven by the residential sector, particularly the usage of chlorine-containing disinfectants. Emission peaks occur during summer, corresponding to the high demand for water disinfection. Lower emissions of $Cl_2$ and HOCl are observed in February.

## 4 Evaluation and Discussion

### 4.1 Evaluation based on model simulations

To verify the accuracy of the chlorine emission inventory, we incorporated the emission data to the chemical transport model CMAQ (Community Multiscale Air Quality Model, version 5.2.1), and compared the simulated concentrations of chlorine species with the observation. The simulation was run for the whole year of 2019, with a prior 10-day as the spin-up time. The meteorological inputs for the CMAQ model were driven by the Weather Research and Forecasting model (WRF version 3.9.1). The meteorological initial and boundary conditions were provided by NCEP/NCAR FNL reanalysis data with a horizontal resolution of 1°×1°. The modelling domain of the WRF and CMAQ model covers mainland China with a horizontal resolution of 36 km. The model has 23 vertical layers and reaches up to the top of 50 hPa. We used SAPRC07TIC as the gas-phase chemical mechanism and AERO6i as the aerosol mechanism in the CMAQ model (Carter, 2010; Hutzell et al., 2012; Xie et al., 2013; Murphy et al., 2017; Pye et al., 2017), which includes comprehensive gaseous reactions of chlorine species and the heterogeneous reaction of $N_2O_5$ and particulate chlorine producing $ClNO_2$ (Sarwar et al., 2012; Sarwar et al., 2014). The chemical initial and boundary conditions of the CMAQ model were provided by the modelling results from the GEOS-Chem model simulation (Zhu et al., 2024; Wang et al., 2022b). For the emission, the anthropogenic emission of conventional pollutants for China was derived from MEIC (Multi-resolution Emission Inventory for China) 2019 (Zheng et al., 2018; Zheng et al., 2021), and those for east Asia (except China) were obtained from the MIX emission inventory (Li et al., 2017). International shipping emissions were obtained from EDGAR (Emissions Database for Global Atmospheric Research v6.1) (Johansson et al., 2017). We also incorporated the anthropogenic chlorine emissions developed in this study to the model. The

biogenic emissions were calculated from the Model of Emissions of Gas and Aerosols from Nature (MEGAN version 2.1) (Guenther et al., 2012) driven by the meteorological inputs from the WRF model.

The field observations of chlorine species are sparse in China. Table 6 presents the comparison of simulated concentrations of HCl, pCl, $Cl_2$, HOCl, and $ClNO_2$ to the observed values in China collected from available literature. It should be noted that the observations are varied in different years (from 2012 to 2022) and regions, and we used the simulated value during the same months at the same location but in the year 2019 to make a rough comparison. Overall, the simulated mean concentrations of HCl and pCl reasonably matched the observations in a similar magnitude, which suggested the faithful estimation of their

emission in this study. For the measurements in some cities (e.g., Wuhan, Guangzhou, Shenzhen, Suzhou, and Beihai) before 2019, the HCl and pCl concentrations were underestimated in the 2019 simulation, which can be attributed to the emission reduction due to the implementation of stringent air quality control in China since 2013. The simulated values of $Cl_2$, HOCl and $ClNO_2$ also fell in the range of the observations, which indicated the reasonable estimations of their emissions and their suitable applications to the chemical transport model.

**4.2 Comparison with previous studies**

We compared our results with the chlorine emissions in China in recent years from other previous studies (Table 5). We regrouped their emission into our source category for a comprehensive comparison. The HCl emission in this study is higher than the emissions in our previous study due to the inclusion of emissions from industrial processes and biomass burning. The

585 HCl emission from coal combustion in this study (2019) is lower than those in 2012 and 2014 due to the reduction of coal consumption in recent years after the implementation of the Clean Air Action in China in 2013. The HCl emission from waste incineration in this study is higher because our previous study only included emissions from MSW incineration stations. The total HCl emission estimated by Fu et al. (2018), Zhang et al. (2022), and Yin et al. (2022) are 458 Gg (2014), 705 Gg (2014), 270 Gg (2019), respectively. Our estimations (361 Gg) are within the range of their results. In this study, HCl emissions from

590 coal combustion are about twice as high as those reported by Fu et al. (2018) and Yin et al. (2022), but significantly lower than the estimation by Zhang et al. (2022). This discrepancy primarily arises from the varied application of emission factors and control technologies. For coal combustion, our study relies on source data derived from on-site observations, differing from Fu et al. (2018) who used control technology application ratios based on national policies, Yin et al. (2022) who employed foreign application ratios, and Zhang et al. (2022) who used an S-curve formula leading to overestimations. Our control

technology application ratios are based on domestic research literature, which is deemed more reasonable. Furthermore, our

study reports substantially lower HCl emissions from waste incineration compared to Fu et al. (2018) and Zhang et al. (2022). This reduction is attributed to advancements in China's MSW management, with greater emphasis on centralized waste collection and reduced open burning compared to the situation in 2014. Additionally, considering the economic disparities, urbanization, and uneven waste treatment technology adoption among provinces, we have adopted a differentiated approach, utilizing waste treatment rates from the Statistical Yearbook to estimate open burning ratios for each province. This approach offers a more nuanced representation of provincial differences compared to the uniform national open burning ratios employed by Yin et al. (2022) and Zhang et al. (2022). In our study, biomass combustion is a notable contributor to HCl emissions, primarily due to disparities in estimations of household combustion rates and outdoor burning methods compared to previous research. For household combustion rate, compared with Yin et al. (2022) who directly adopted half of Zhou et al. (2017)'s research, we estimated the household combustion rate for each province according to different linear trends based on the study of Liu et al. (2022), resulting in a more detailed estimation. Our approach to open burning employs a bottom-up emission factor method, mitigating potential underestimations resulting from satellite-based detection, which may overlook small-scale or short-term fire events. Moreover, our estimate of HCl emissions in the Yangtze River Delta (YRD) region is approximately 1.5 times that reported by Yi et al. (2021), who underestimated HCl emissions due to their exclusive focus on centralized coal combustion, neglecting dispersed coal combustion. In contrast, our study not only encompasses urban waste incineration facilities but also considers emissions from county towns and open burning.

The total emission of fine particulate Cl⁻ in this study is within the range of the emissions reported by Fu et al. (2018), Zhang et al. (2022), and Yin et al. (2022). All of these studies highlight biomass burning as the major contributor (74%, 75%, 57%, and 78% in respective studies). The emissions from coal combustion, industrial processes, and biomass burning are generally consistent across these studies. Regarding waste incineration, emissions reported by Fu et al. (2018) and Zhang et al. (2022) are notably higher than those in our study and Yin et al. (2022). This difference can be attributed to significant improvements in China's municipal solid waste management since 2014, with a greater emphasis on centralized waste collection and reduced open burning, leading to decreased emissions. Yin et al. (2022), Qiu et al. (2019b) and this study all considered emissions from cooking. In this study, we employed lower flue gas flow rates and shorter cooking durations following national standards and actual conditions, and lower proportions of Cl⁻ in $PM_{2.5}$ based on the local measured data from the literature, which reduced the cooking emission compared with the other two studies. The estimated emissions from the YRD in this study are higher than those from Yin et al. (2022) because our activity data is considered more comprehensively.

The total emission of $Cl_2$ in this study is higher than our previous studies because we included additional emissions from usages of disinfectants and pesticides. Yin et al. (2022) and our studies both demonstrated the emission from usages of chlorine-containing disinfectants is as important as those from coal combustion. However, the total $Cl_2$ emission in this study is ~2 times higher than those estimated in the study of Yin et al. (2022). In this study, emissions from coal combustion are roughly twice as high as those in Yin et al. (2022) due to the use of locally measured emission factors and the application ratio of domestic control technologies. The emissions from disinfectant use in this study are about three times higher than those in Yin et al. (2022). This increase is attributed to the expanded sources of emissions considered in this study, as well as improvements in the calculation methods for chlorine addition in swimming pools. Firstly, this study includes emissions from county-level water treatment, county-level wastewater treatment, tap water usage, and environmental disinfection processes, accounting for 5%, 6%, 2%, and 13% of the total emissions from disinfectant use, respectively. These sources are not negligible. Secondly, localized improvements were made to some emission factors. Regarding the chlorine addition in swimming pools, while Yin et al. (2022) used a concentration of 1 mg $L^{-1}$ from national standards, this study cited experimental research literature and used outdoor and indoor chlorine addition rates of 1.89 mg $L^{-1}$ and 1.19 mg $L^{-1}$ (Wong et al., 2017; Wang et al., 2002) (https://bg.qianzhan.com/report/detail/300/210803-5009dc2a.html, last access: 1 August 2024), respectively. Lastly, we improved the estimation method for swimming pool emissions by distinguishing between the open times of indoor and outdoor pools. Unlike the assumption in Yin et al. (2022) that all pools are open in the summer, this study assumes year-round operation for indoor pools and only summer operation for outdoor pools, which is a more realistic scenario. This led to increased disinfectant use in swimming pools, resulting in higher emissions compared to Yin et al. (2022). The estimated emissions from the YRD in this study are higher than those from Yi et al. (2022) because we consider dispersed coal combustion during coal combustion, and activity data such as water treatment, waste water treatment, and environmental disinfection also consider rural areas. The estimated emissions in Shanghai are half that of Li et al. (2020) because their main source of emissions is overestimated from cooling towers, which was also mentioned in the study of Yin et al. (2022).

The research on the development of HOCl emission inventory is limited. This study and Yin et al. (2022) both showed that the emission from the usage of chlorine-containing disinfectants is the major contributor to HOCl emission. However, we estimated the HOCl emission to be ~ 3 times higher than those estimated by Yin et al. (2022). As mentioned above, this higher estimation can be attributed to the addition of emissions from swimming pools, environmental disinfection, tap water use, and pesticides.

**4.3 Uncertainty and limitation**

We applied the Monte Carlo method to quantify the uncertainties of the ACEIC inventory. Normal distributions with coefficients of variation (CV) ranging from 5% to 50% were assumed for activity data according to previous studies (Yi et al., 2021; Li et al., 2020b; Zheng et al., 2022; Fu et al., 2018). For the emission factors, probability distributions were fitted for parameters with adequate measurement data. For parameters with limited measurement data, probability distributions were assumed as uniform or log-normal distributions. The detailed uncertainty assumptions for the activity data and emission factors are summarized in Tables S18 and S19. The uncertainties for HCl, pCl, $Cl_2$, and HOCl emissions were estimated at a 95% confidence interval, resulting in percentage ranges of -18% to 27%, -27% to 59%, -10% to 15%, and -12% to 18%, respectively (Figure 7). It can be seen that the estimated emissions of HCl and pCl are within the uncertainty ranges of other studies. Due to the additional sources of $Cl_2$ and HOCl in this study, the emissions are relatively higher compared with Yin et al. (2022). However, the percentage of uncertainty for all chlorine species generally reduces in this study compared with the other studies.

The potential limitations of the ACEIC emission inventory that require further refinement are summarized as follows. (1) Limited sources for some activity data and emission factors: For example, for estimating chlorine emissions from hospital disinfection, the average disinfectant usage derived from five hospitals may not accurately reflect the variability in usage across different regions and facility types. There is a need for further localized investigation on activity data and experimental studies on emission factors. This is particularly important for sectors such as waste treatment and usage of chlorine-containing disinfectant, where more localized observation data is needed to improve the accuracy of emission estimates. (2) Complexity of indoor-outdoor air exchange: The study simplifies the treatment of some indoor chlorine emissions (e.g., environmental disinfection) by assuming rapid air exchange between indoor and outdoor environments. However, the actual exchange rate can vary significantly due to differences in building structures, ventilation systems, and climatic conditions. This variability can lead to uncertainties in estimating the impact of indoor emissions on outdoor air quality. Future research should incorporate more detailed assessments of these factors to improve the accuracy of emission estimates. (3) Spatial and temporal distribution of emissions: The study employs generalized proxies, such as population distribution, for spatial allocation of emissions from point sources like residential coal combustion and biomass burning. This approach may not accurately capture the localized nature of emissions and their impacts. Additionally, the temporal allocation of pesticide emissions based on monthly production does not fully account for regional variations in crop types and climate. More precise data on spatial and temporal allocation factors, such as detailed information on the operating scales of point sources, are needed to reduce uncertainties and enhance the accuracy of the inventory. (4) Inclusion of emissions from additional anthropogenic sources: The inventory can be refined by including emissions from other anthropogenic activities that release chlorine species. For example, public places

such as hotels, schools, and shopping malls undergo daily disinfection, which can lead to the release of a large amount of $Cl_2$ and HOCl. Despite the limitations outlined, the modeling results with this emission inventory results align with the observation, suggesting that the methodologies used provide a reasonable approximation of actual conditions. The study's acknowledgment of these limitations and its commitment to addressing them through future research indicate a rigorous and transparent approach. Continued refinement and incorporation of more detailed data are expected to enhance the inventory's accuracy and reliability.

## 5 Conclusion and implication

In this study, we developed the anthropogenic chlorine emission inventory for China (ACEIC 2019) using emission factors mainly based on local measurements and survey data from the literature. This inventory includes 4 chlorine species (HCl, fine particle $Cl^-$, $Cl_2$, and HOCl) and 41 specific anthropogenic sources. We estimate that the total emissions of HCl, fine particle $Cl^-$, $Cl_2$, and HOCl in mainland China in 2019 were 361 (-18%~27%), 174 (-27%~59%), 18 (-10%~15%) and 79 (-12%~18%) Gg, respectively. This emission inventory was incorporated into a chemical transport model and the simulated concentrations of chlorine species agreed reasonably with the observation, suggesting the relatively faithful estimation of their emissions.

In terms of energy consumption sources, for HCl emissions, coal combustion and biomass combustion are the main sources, accounting for 47% and 38% of the total, respectively. For fine particle $Cl^-$, biomass combustion is the main contributor (74%) to the total emissions. For $Cl_2$, it mainly comes from the usage of chlorine-containing disinfectants (59%) and coal burning (41%). For HOCl emissions, the use of chlorine-containing disinfectants is the main reason (99%). Among them, waste water treatment (43%) and water treatment (31%) are the human activities with the highest emissions during the usage of chlorine-containing disinfectants.

From the perspective of the economic sector, HCl emissions mainly come from the industrial (43%) and biomass combustion (38%) sectors. Biomass combustion and the industrial sector are the main sources of fine particulate chlorine, accounting for 74% and 19% of total emissions, respectively. $Cl_2$ emissions mainly come from the residential and industrial sectors, accounting for 61% and 29% of the total emissions, respectively. HOCl emissions mainly come from the residential (90%) sector. The use of chlorine-containing disinfectants is the most important source of $Cl_2$ and HOCl in the residential sector.

From the perspective of provincial emissions, Heilongjiang has the highest emissions of HCl (36.76 Gg) and fine particle Cl⁻ (27.18 Gg). Guangdong has the highest emissions of $Cl_2$ (1.59 Gg) and HOCl (9.92 Gg). High HCl and fine particle Cl⁻ emission areas are located in the North China Plain, Northeast China, Central China, and Yangtze River Delta, while $Cl_2$ and HOCl emission areas are located in the Pearl River Delta, Yangtze River Delta, and Beijing-Tianjin-Hebei region. In terms of monthly changes, due to active agricultural activities, the emissions of HCl and fine particle Cl⁻ are relatively high during early spring (February to April) and winter (December to January), while the emissions of $Cl_2$ and HOCl are higher in summer due to the high demand for corresponding disinfection water.

The results of this study demonstrate that chlorine-containing disinfectants are significant sources of $Cl_2$ and HOCl. The demand for these disinfectants has risen substantially in recent years due to the spread of COVID-19 worldwide. It is anticipated that the increase in chlorine emissions following the outbreak of COVID-19 will play a more important role in tropospheric chemistry. Furthermore, most studies focusing on emissions for a specific year and the long-term trends of chlorine emissions remain unknown. Therefore, an inter-annual emission inventory of chlorine species is essential for controlling emissions. It is worth noting that $Cl_2$ and HOCl emissions typically peak during the summer when severe $O_3$ pollution frequently occurs in many parts of China. Therefore, future investigations should assess the impact of chlorine emissions on summertime $O_3$ formation. This inventory provides valuable insights into the anthropogenic sources of chlorine species and is conducive to the development of emission control strategies to mitigate secondary pollution in China. We suggest that air quality modeling studies include the chlorine emission inventory to accurately simulate tropospheric chlorine chemistry.

**Author contributions**

Y.M.L. and Q.F. initiated the research. Y.M.L. designed the research framework. S.T.L. and Y.M.L. collected the materials. S.T.L. calculated the emissions and drew the figures. Y.M.L. and S.T.L. analyzed the results and wrote the paper with inputs from all authors. All authors contributed to the discussion and improvement of the paper.

**Financial support**

This research has been supported by the Key-Area Research and Development Program of Guangdong Province (2020B1111360003), Guangdong Major Project of Basic and Applied Basic Research (2020B0301030004), Science and Technology Program of Guangdong Province (Science and Technology Innovation Platform Category) (2019B121201002),

and the National Natural Science Foundation of China (42105097), and Guangdong Basic and Applied Basic Research Foundation (2023A1515010162).

**Competing interests**

The authors declare that they have no conflict of interest.

**Code/Data availability**

The code or data used in this study are available upon request from Yiming Liu (liuym88@mail.sysu.edu.cn) and Siting Li (list23@mail2.sysu.edu.cn).

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

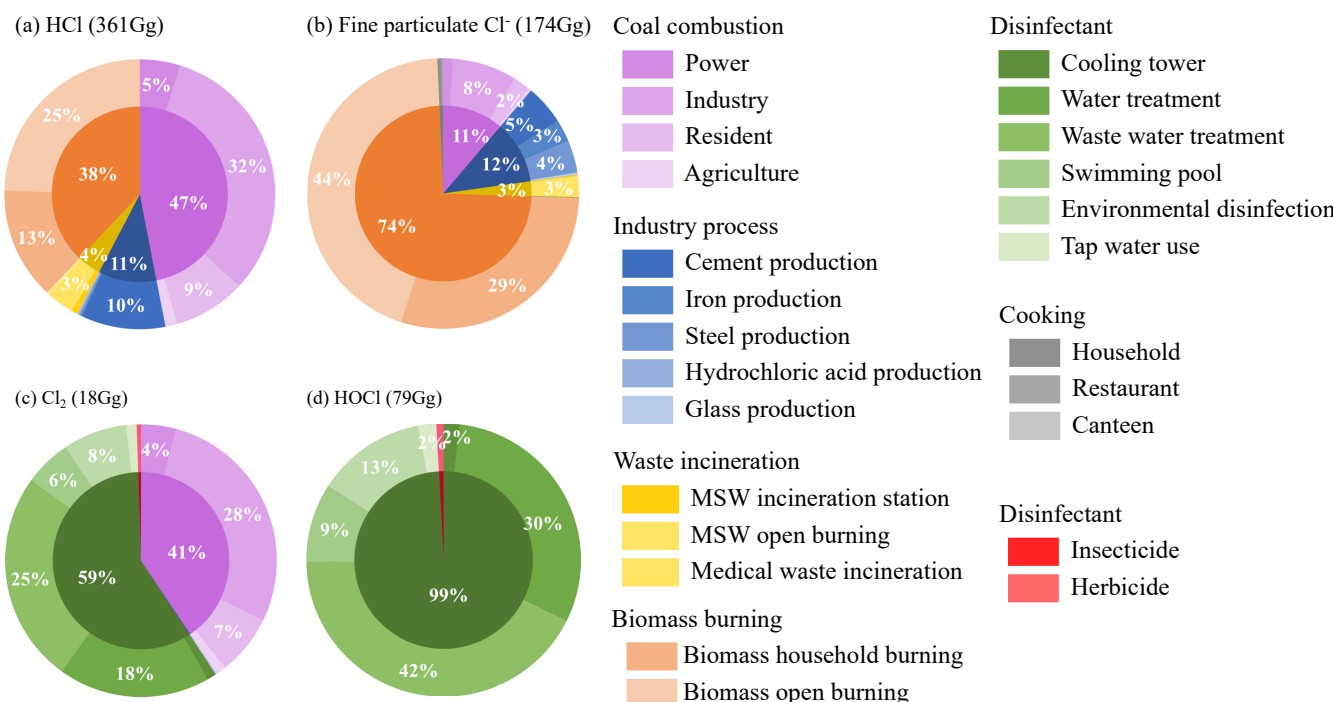

(a) HCl (361Gg)  (b) Fine particulate Cl⁻ (174Gg)

Coal combustion
- Power
- Industry
- Resident
- Agriculture

Industry process
- Cement production
- Iron production
- Steel production
- Hydrochloric acid production
- Glass production

Waste incineration
- MSW incineration station
- MSW open burning
- Medical waste incineration

Biomass burning
- Biomass household burning
- Biomass open burning

Disinfectant
- Cooling tower
- Water treatment
- Waste water treatment
- Swimming pool
- Environmental disinfection
- Tap water use

Cooking
- Household
- Restaurant
- Canteen

Disinfectant
- Insecticide
- Herbicide

Figure 1 Contributions of different source categories to the anthropogenic emissions of HCl (a), fine particulate Cl⁻ (b), Cl₂ (c), and HOCl (d) in 2019. The aggregation of 7 primary source categories from 41 specific sources can be found in Table S1.

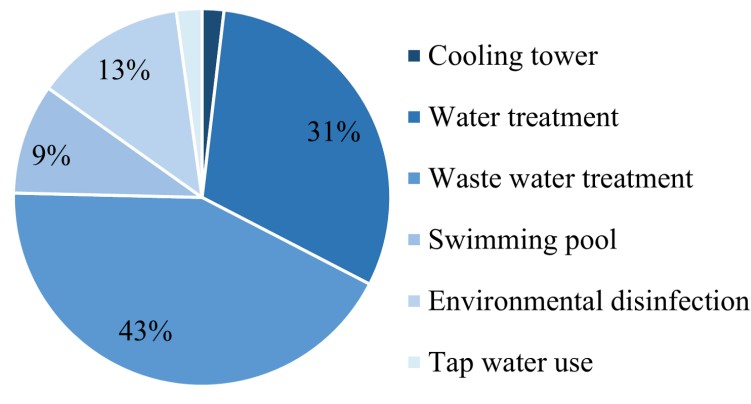

Figure 2 Proportion of chlorine emissions from different usages of chlorine-containing disinfectant in China.

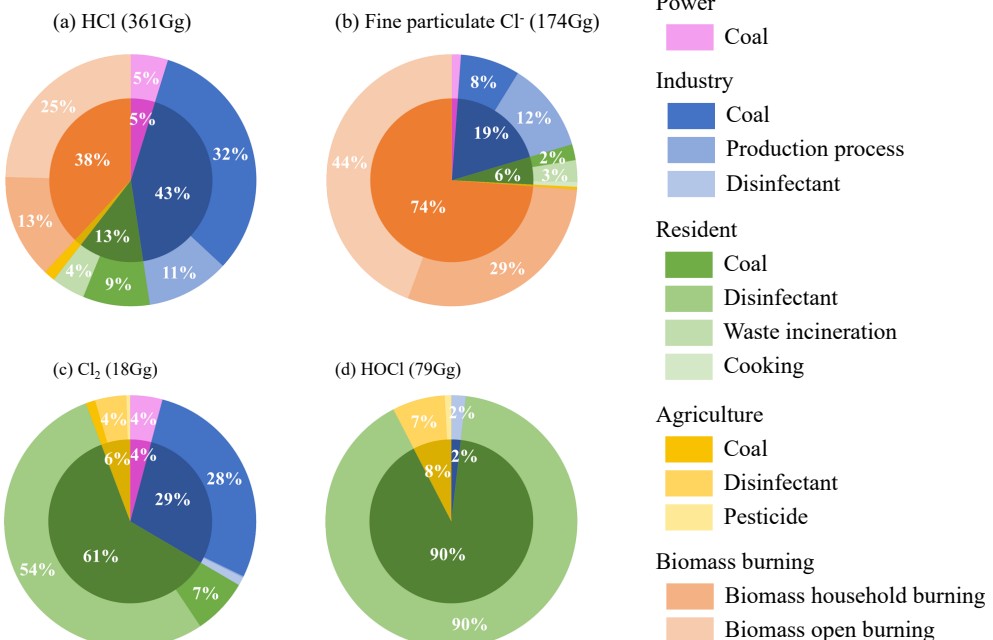

Figure 3 Contributions of different economic sectors to the anthropogenic emissions of HCl (a), fine particulate Cl⁻ (b), Cl₂ (c), and HOCl (d) in 2019. The aggregation of 5 economic sectors from 41 specific sources can be found in Table S2.

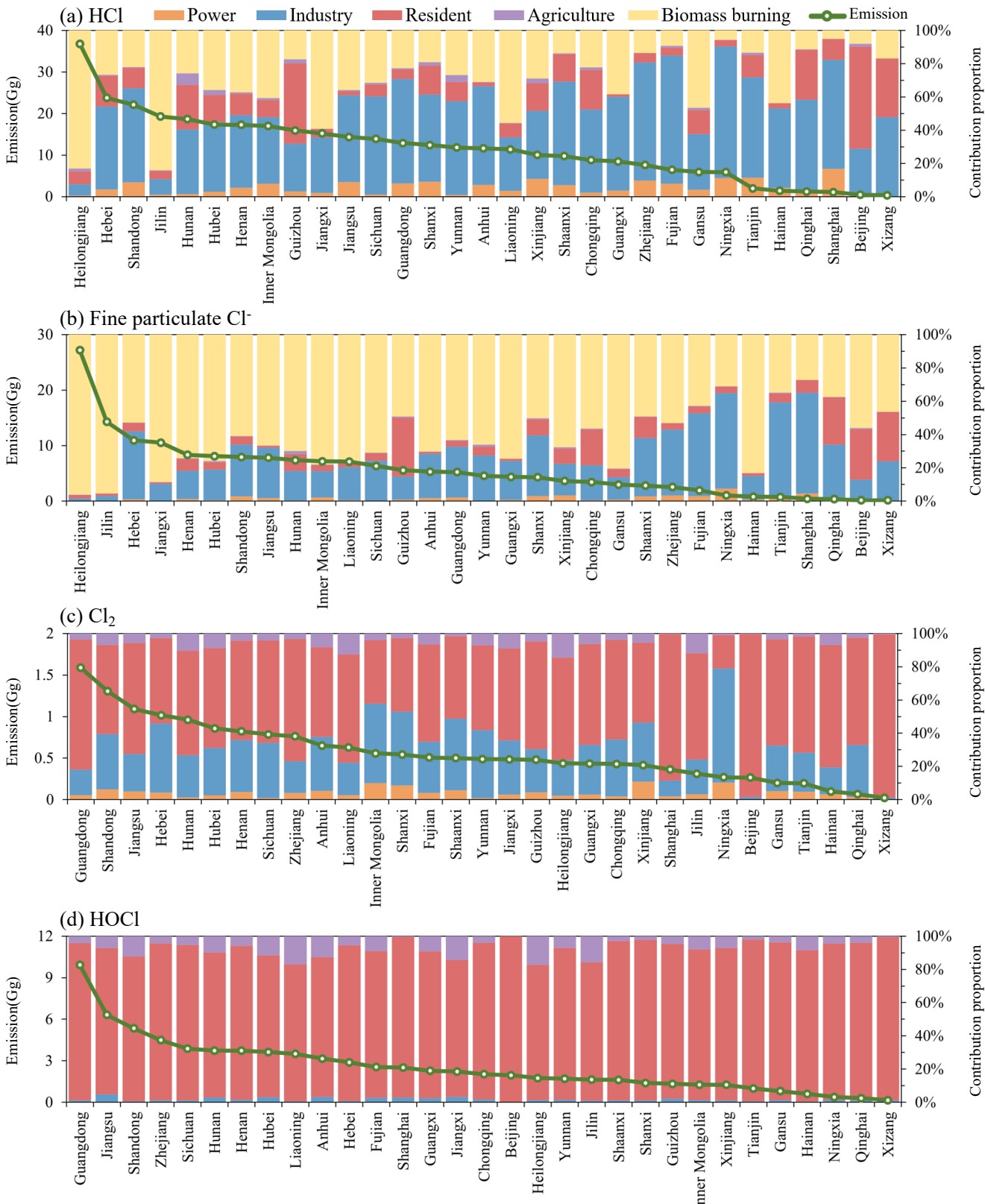

Figure 4 Emissions (green line) and contribution proportions of HCl (a), fine particulate Cl⁻ (b), Cl₂ (c), and HOCl (d) by province in 2019.

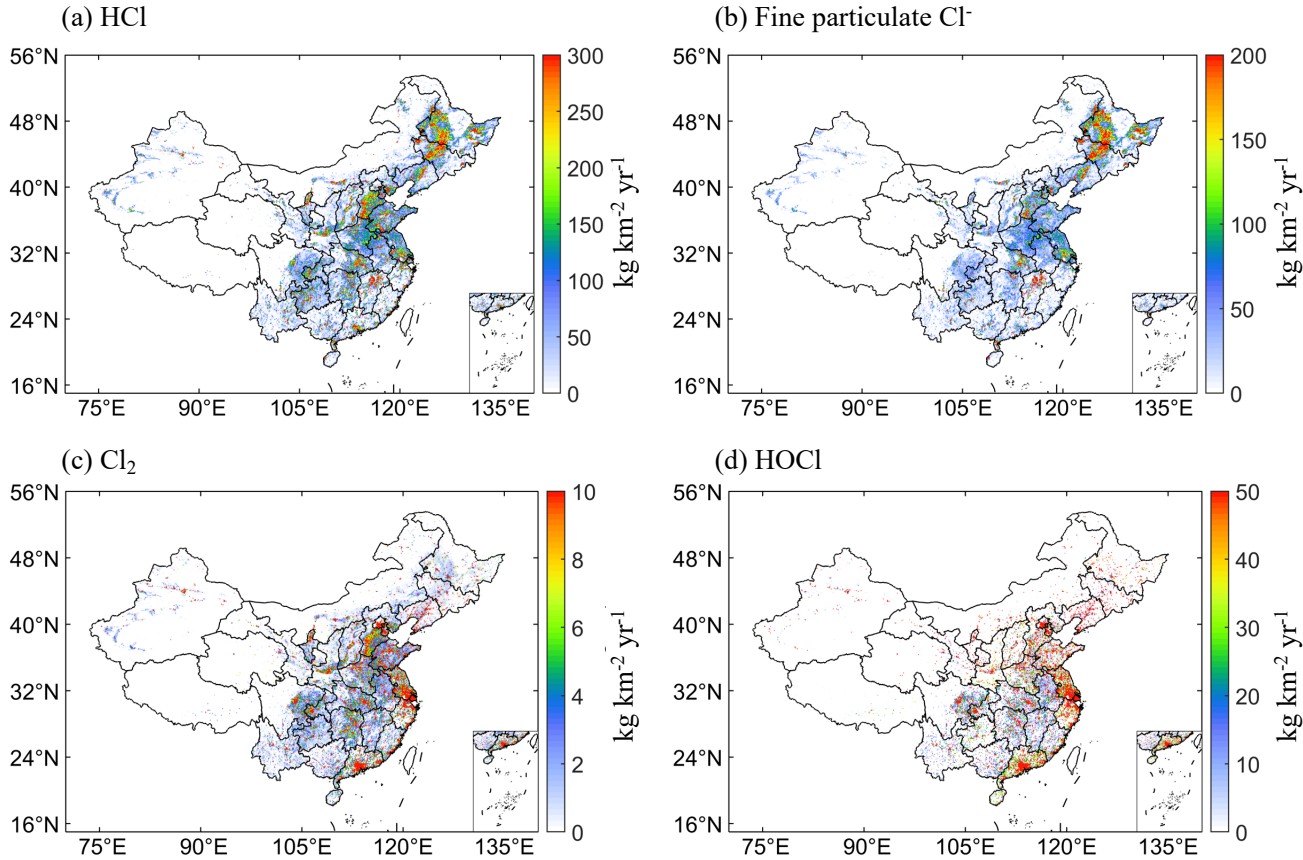

Figure 5 Spatial distribution of anthropogenic HCl, fine particulate Cl⁻, Cl₂, and HOCl emissions in 2019 at 0.1° × 0.1° resolution.

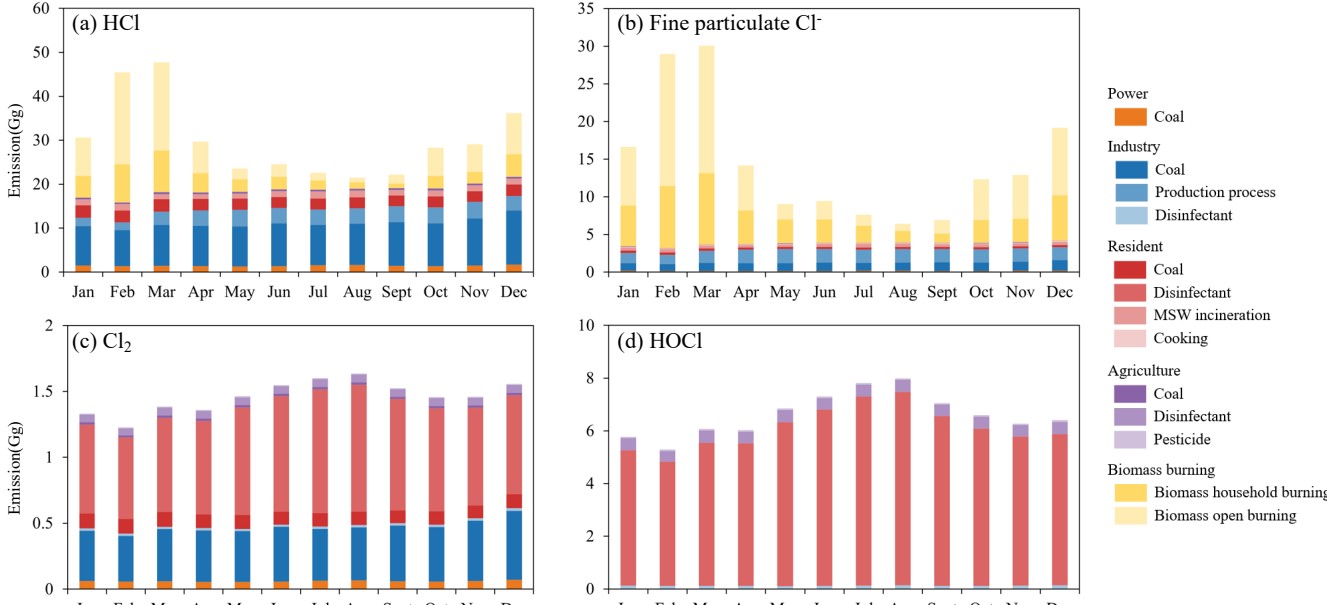

Figure 6 Monthly variation of anthropogenic HCl (a), fine particulate Cl⁻ (b), Cl₂ (c), and HOCl (d) emissions by economic sector in 2019.

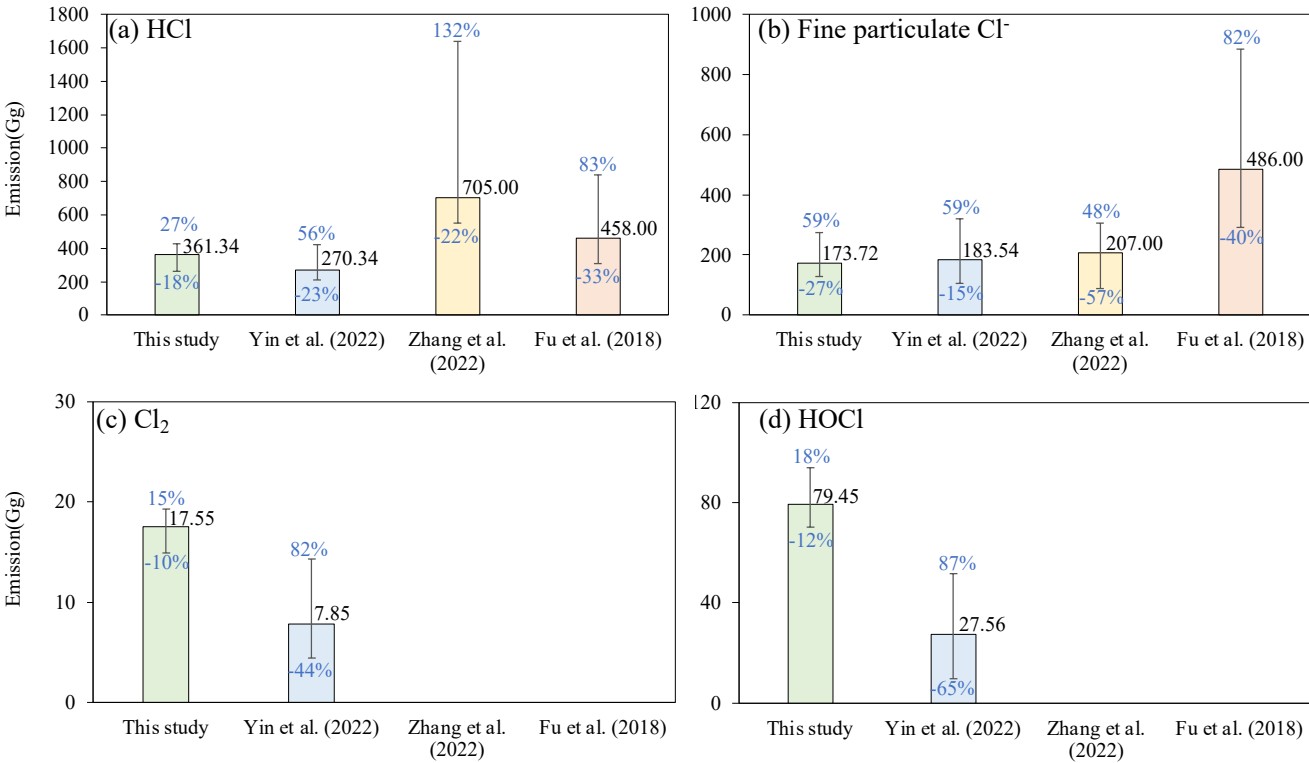

Figure7 Comparison of anthropogenic chlorine emissions and uncertainty ranges (blue text) with other studies.

Table 1 General information about the ACEIC inventory

| Item | Information |
| --- | --- |
| Name | Anthropogenic chlorine emissions for China (ACEIC) |
| Domain | 31 provinces in mainland China |
| Species | HCl, fine particulate Cl$^-$, Cl$_2$, and HOCl |
| Source categories | 7 major categories: (1) coal combustion, (2) industrial production process, (3) waste incineration, (4) biomass burning, (5) cooking, (6) usages of chlorine-containing disinfection, and (7) usages of pesticide |
| Base year | 2019 |
| Spatial resolution | 0.1°×0.1° |
| Temporal resolution | Monthly |

Table 2 Anthropogenic chlorine emission in China by source category in 2019.

| Source category | Sub-category | Emission (Gg) | | | |
|---|---|---|---|---|---|
| | | HCl | pCl | $Cl_2$ | HOCl |
| Coal combustion | Power | 17.31 | 2.03 | 0.72 | |
| | Industrial | 115.98 | 13.29 | 4.92 | |
| | Residential | 31.11 | 3.54 | 1.26 | |
| | Agriculture | 5.30 | 0.60 | 0.21 | |
| | Sum of coal combustion | 169.69 | 19.46 | 7.12 | 0.00 |
| Industrial production process | Cement production | 36.31 | 8.20 | | |
| | Iron production | 0.46 | 4.66 | | |
| | Steel production | 0.76 | 6.75 | | |
| | HCl production | 0.56 | | | |
| | Flat glass production | 0.56 | 0.49 | 0.03 | |
| | Sum of industrial production process | 38.65 | 20.10 | 0.03 | 0.00 |
| Waste incineration | MSW incineration station | 2.59 | 0.20 | | |
| | MSW open burning | 11.74 | 4.44 | | |
| | Medical waste incineration | 1.33 | 0.10 | | |
| | Sum of waste incineration | 15.66 | 4.74 | 0.00 | 0.00 |
| Biomass burning | Household burning | 48.43 | 51.23 | | |
| | Open burning | 88.90 | 76.99 | | |
| | Sum of biomass burning | 137.34 | 128.22 | 0.00 | 0.00 |
| Cooking | Household | | 0.74 | | |
| | Restaurant | | 0.40 | | |
| | Canteen | | 0.05 | | |
| | Sum of cooking | 0.00 | 1.19 | 0.00 | 0.00 |
| Disinfectant | Cooling tower | | | 0.19 | 1.47 |
| | Water treatment | | | 3.17 | 24.21 |
| | Waste water treatment | | | 4.41 | 33.70 |
| | Swimming pool | | | 0.98 | 7.45 |
| | Environment disinfectant | | | 1.34 | 10.23 |
| | Tap water use | | | 0.23 | 1.73 |
| | Sum of disinfectant | 0.00 | 0.00 | 10.32 | 78.78 |
| Pesticide | Insecticide | | | 0.09 | 0.65 |
| | Herbicide | | | 0.00 | 0.02 |
| | Sum of pesticide | 0.00 | 0.00 | 0.09 | 0.67 |
| Sum of all categories | | 361.34 | 173.72 | 17.55 | 79.45 |

Table 3 Anthropogenic chlorine emission in China by economic sector in 2019.

| Sector | Subsector | Emission (Gg) | | | |
|---|---|---|---|---|---|
| | | HCl | pCl | $Cl_2$ | HOCl |
| Power | Coal combustion | 17.31 | 2.03 | 0.73 | |
| Industry | Industrial coal combustion | 115.98 | 13.29 | 4.92 | |
| | Industrial production process | 38.65 | 20.10 | 0.03 | |
| | Industrial usage of disinfectant | 0.00 | 0.00 | 0.19 | 1.47 |
| | Sum of industry | 154.62 | 33.39 | 5.14 | 1.47 |
| Residential | Residential coal combustion | 31.11 | 3.54 | 1.26 | |
| | Residential usage of disinfectant | | | 9.42 | 71.93 |
| | Waste incineration | 15.66 | 4.74 | | |
| | Cooking | | 1.19 | | |
| | Sum of residential sector | 46.77 | 9.47 | 10.68 | 71.93 |
| Agriculture | Agricultural coal combustion | 5.30 | 0.60 | 0.21 | |
| | Agricultural usage of disinfectant | | | 0.70 | 5.38 |
| | Agricultural usage of pesticide | | | 0.09 | 0.67 |
| | Sum of agriculture | 5.30 | 0.60 | 1.01 | 6.05 |
| Biomass burning | Biomass household burning | 48.43 | 51.23 | | |
| | Biomass open burning | 88.90 | 76.99 | | |
| | Sum of biomass burning | 137.34 | 128.22 | | |
| Sum of all sectors | | 361.34 | 173.72 | 17.55 | 79.45 |

Table 4 Anthropogenic chlorine emissions by province in 2019.

| Province | Emission (Gg) | | | |
|---|---|---|---|---|
| | HCl | pCl | Cl$_2$ | HOCl |
| Beijing | 0.46 | 0.16 | 0.27 | 1.95 |
| Tianjin | 2.00 | 0.77 | 0.20 | 1.01 |
| Hebei | 23.76 | 10.97 | 1.02 | 2.89 |
| Shanxi | 12.42 | 4.29 | 0.54 | 1.40 |
| Inner Mongolia | 17.00 | 7.17 | 0.56 | 1.28 |
| Liaoning | 11.38 | 7.13 | 0.63 | 3.50 |
| Jilin | 19.30 | 14.29 | 0.31 | 1.63 |
| Heilongjiang | 36.76 | 27.18 | 0.44 | 1.75 |
| Shanghai | 1.09 | 0.46 | 0.36 | 2.51 |
| Jiangsu | 14.37 | 7.83 | 1.09 | 6.32 |
| Zhejiang | 7.61 | 2.54 | 0.76 | 4.49 |
| Anhui | 11.61 | 5.32 | 0.65 | 3.16 |
| Fujian | 6.45 | 1.96 | 0.51 | 2.55 |
| Jiangxi | 15.22 | 10.53 | 0.49 | 2.23 |
| Shandong | 22.12 | 7.94 | 1.31 | 5.34 |
| Henan | 17.28 | 8.35 | 0.82 | 3.73 |
| Hubei | 17.37 | 8.11 | 0.86 | 3.63 |
| Hunan | 18.65 | 7.38 | 0.96 | 3.74 |
| Guangdong | 12.89 | 5.25 | 1.59 | 9.92 |
| Guangxi | 8.50 | 4.38 | 0.43 | 2.28 |
| Hainan | 1.43 | 0.81 | 0.10 | 0.62 |
| Chongqing | 8.81 | 3.45 | 0.43 | 2.03 |
| Sichuan | 13.92 | 6.35 | 0.79 | 3.87 |
| Guizhou | 15.93 | 5.56 | 0.48 | 1.34 |
| Yunnan | 11.82 | 4.57 | 0.49 | 1.71 |
| Xizang | 0.35 | 0.16 | 0.02 | 0.16 |
| Shaanxi | 9.76 | 2.80 | 0.50 | 1.63 |
| Gansu | 5.93 | 2.97 | 0.20 | 0.81 |
| Qinghai | 1.25 | 0.38 | 0.07 | 0.30 |
| Ningxia | 5.89 | 1.04 | 0.27 | 0.38 |
| Xinjiang | 10.03 | 3.62 | 0.42 | 1.26 |
| Sum | 361.34 | 173.72 | 17.55 | 79.45 |

Table 5 Comparison of anthropogenic chlorine emissions with other studies. The values in brackets are the proportion of different source categories.

| Species | Study | Year | Total | Coal combustion | Industrial process | Waste incineration | Biomass burning | Cooking | Usage of disinfectant | Usage of pesticide |
|---|---|---|---|---|---|---|---|---|---|---|
| HCl (Gg) | This study | 2019 | 361.34 | 169.69 (47%) | 38.65 (11%) | 15.66 (4%) | 137.34 (38%) | | | |
| | Liu et al. (2018) | 2012 | 235.80 | 232.90 (99%) | | 2.90 (1%) | | | | |
| | Hong et al. (2020) | 2014 | 223.40 | 219.20 (98%) | | 4.20 (2%) | | | | |
| | Fu et al. (2018) [a] | 2014 | 458.00 | 87.00 (19%) | 36.60 (8%) | 187.80 (41%) | 146.60 (32%) | | | |
| | Zhang et al. (2022) [a] | 2014 | 705.00 | 310.2 (44%) | 21.15 (3%) | 338.40 (48%) | 35.25 (5%) | | | |
| | Yin et al. (2022) | 2019 | 270.34 | 80.61 (30%) | 57.09 (21%) | 35.01 (13%) | 97.63 (36%) | | | |
| pCl (Gg) | This study | 2019 | 173.72 | 19.46 (11%) | 20.10 (12%) | 4.74 (3%) | 128.22 (74%) | 1.19 (1%) | | |
| | Fu et al. (2018) [a] | 2014 | 486.00 | 24.30 (5%) | 29.20 (6%) | 68.00 (14%) | 364.50 (75%) | | | |
| | Zhang et al. (2022) [a] | 2014 | 207.00 | 20.70 (10%) | 16.56 (8%) | 51.75 (25%) | 117.99 (57%) | | | |
| | Yin et al. (2022) | 2019 | 183.54 | 11.86 (6%) | 11.88 (6%) | 9.78 (5%) | 143.19 (78%) | 6.85 (4%) | | |
| Cl₂ (Gg) | This study | 2019 | 17.55 | 7.12 (41%) | 0.03 (0%) | | | | 10.32 (59%) | 0.09 (0%) |
| | Liu et al. (2018) | 2012 | 9.40 | 9.40 (100%) | | | | | | |
| | Hong et al. (2020) | 2014 | 8.90 | 8.90 (100%) | | | | | | |
| | Yin et al. (2022) | 2019 | 7.85 | 3.46 (44%) | 0.78 (10%) | | | | 3.61 (46%) | |
| HOCl (Gg) | This study | 2019 | 79.45 | | | | | | 78.78 (99%) | 0.67 (1%) |
| | Yin et al. (2022) | 2019 | 27.56 | | | | | | 27.56 (100%) | |

[a] The emissions from different source categories are estimated by multiplying the total emission with the corresponding proportions reported in the literature.

Table 6 Comparison between simulated and observed concentrations of chlorine species. The simulated concentrations are the values during the same measurement period but in the year 2019.

| Species | Location | Period | Average observed concentration (1 min max) | Average simulated concentration (1 h max) | Reference |
|---|---|---|---|---|---|
| HCl ($\mu g\ m^{-3}$) | Beijing | Feb.-Mar., 2015 | 0.45 | 0.47 | Zhang et al. (2018) |
| | Beijing | May-Jun., 2016 | 0.83 | 0.34 | Le Breton et al. (2018) |
| | Nanjing | May, 2017 | 0.4 | 0.77 | Gao et al. (2019a) |
| | Shanghai | Oct.-Nov., 2012 | 0.55 | 1.78 | Shi et al. (2014) |
| pCl ($\mu g\ m^{-3}$) | Shanghai | Oct.-Dec., 2019 | 0.4 | 0.27 | Li et al. (2023) |
| | Beijing | Nov., 2018-Jan., 2019 | 0.4 | 0.74 | Li et al. (2020a) |
| | Yanshan | Nov., 2018-Jan., 2019 | 0.4 | 0.18 | Li et al. (2020a) |
| | Wuhan | Jan., 2018 | 1.5 | 1.35 | Gao et al. (2019b) |
| | Guangzhou | Jun.-Sept., 2013 | 0.35 | 0.09 | Jia et al. (2018) |
| | Shenzhen | Sept.-Oct., 2019 | 0.552 | 0.218 | Wang et al. (2022a) |
| | Shenzhen | 2015 | 0.3 | 0.16 | Wu et al. (2020) |
| | Suzhou | Apr., Aug., Oct., Dec., 2015 | 1.54 | 0.39 | Wang et al. (2016) |
| | Beihai | Mar.-Apr., 2015 | 0.28 | 0.16 | Zhou et al. (2018b) |
| $Cl_2$ (ppt) | Beijing | Oct., 2021-Mar., 2022 | 35 (242) | 27.52 (189.94) | Ma et al. (2023) |
| | Beijing | Dec., 2017 | 40 | 11.63 | Peng et al. (2021) |
| | Changzhou | May-Jun., 2019 | 26 (520) | 8.87 (51.16) | Li et al. (2023) |
| | Shanghai | Oct.-Dec., 2019 | 24 (1100) | 25.73 (158.27) | Li et al. (2023) |
| HOCl (ppt) | Nanjing | Apr., 2018 | 0~600 | 76.96 (358.70) | Xia et al. (2020) |
| $ClNO_2$ (ppt) | Beijing | Dec., 2017 | 70 | 69.93 | Peng et al. (2021) |
| | Taishan | Jul.-Aug., 2014 | 54 (2065) | 68.26 (420.76) | Wang et al. (2017c) |
| | Wangdu | Jun.-Jul., 2014 | 550 (2070) | 145.76 (1092.38) | Tham et al. (2016) |
| | Jinan | Aug.-Sept., 2014 | 94 (776) | 103.02 (686.46) | Wang et al. (2017a) |
| | Beijing | Jun., 2017 | 174.3 (1440) | 89.73 (662.24) | Zhou et al. (2018a) |
| | Changzhou | May-Jun., 2019 | 150 (1300) | 129.28 (1006.57) | Li et al. (2023) |
| | Shanghai | Oct.-Dec., 2019 | 94 (5700) | 148.08 (1039.94) | Li et al. (2023) |
| | Shanghai | Oct.-Nov., 2020 | 50 (400) | 166.73 (1039.94) | Lou et al. (2022) |