# Peer review of "ACEIC: a comprehensive anthropogenic chlorine emission inventory for China"

_EGUsphere, 2023_

## Referee Comment (RC2)

**Review comments on the manuscript titled "ACEIC: a comprehensive anthropogenic chlorine emission inventory for China"**

The chlorine radical assumes a pivotal role in atmospheric chemistry, exerting a significant influence on atmospheric oxidation capacity, thereby contributing to secondary air pollution. Chlorine emanates from diverse sources, encompassing both anthropogenic and sea salt aerosol. While preceding studies have partially addressed certain aspects of chlorine emissions, a more comprehensive and reliable dataset consisting complete chlorine species and their respective source contributions remains a pressing necessity. The present study developed an emission inventory detailing anthropogenic chlorine sources, including HCl, $Cl_2$, pCl and HOCl, across 7 distinct source sectors in China for the year 2018 by using the emission factor methodology. Although the topic is both interesting and significant, the manuscript appears to be hindered by a lack of novelty. Several existing papers have already reported chlorine emissions, spanning cities, regions and even the entire nation of China, encompassing a multitude of sources. The current study closely aligns with these prior works, employing nearly identical methods, activity data and reported emissions factors. Consequently, the findings and uncertainties presented within this study are largely consistent with those from earlier investigations. Therefore, the identification of genuinely novel insights or contributions to the scientific community proves to be a challenge.

**Major concerns:**

1. The primary objective of this study is to establish an extensive inventory encompassing anthropogenic chlorine emissions in China. This inventory includes HCl, fine particle $Cl^-$, $Cl_2$, and HOCl from 7 anthropogenic sources for year 2018. Nonetheless, the clarity in communicating the novelty and distinctiveness of this study from its precursors is lacking. Several papers have already reported on China's chlorine emissions inventory for various years, such as 2014 (Fu et al., 2018), 2019 (Yin et al., 2022), and even the range spanning 1960 to 2014 (Zhang et al., 2022). Particularly, the work by Yin (2022) has extensively detailed emissions of HCl, pCl, $Cl_2$, and HOCl, with source categories covering 22 sectors. Notably, they employed a spatial resolution of $0.1° × 0.1°$ and reported the temporal resolution as well. The outcomes of the present study remain consistent with those presented in previous investigations (Fu et al., 2018; Zhang et al., 2022; Yin et al., 2022), and the level of uncertainties closely mirrors that of earlier researches, as indicated in Table 6. It is pertinent to highlight that this study adopts a methodology akin to prior works, employing the emission factor method, which does raise concerns about potential duplication of prior research efforts.

2. The authors have highlighted that there exists uncertainty regarding previous anthropogenic sources, necessitating further investigation. However, the specific nature of these uncertainties and the areas requiring further inquiry remain unclear.

Has the current research succeeded in mitigating any of these uncertainties? If so, through what means has this reduction been achieved? Have the authors incorporated more precise activity data, or have they embraced empirically measured emission factor data? It appears that the authors predominantly relied on activity data from statistical yearbooks and incorporated emissions factor information gleaned from previous literature. Consequently, the uncertainties associated with HCl, pCl, $Cl_2$, and HOCl emissions spanned a range of -48% to 45%, -59% to 89%, -44% to 58%, and -44% to 79% respectively. These figures closely mirror those from earlier studies, as also evidenced in Table 6.

3. The authors emphasized that "it should be noted that some important sources of chlorine emissions have been overlooked, leading to large uncertainties in recent decade estimates (219-707 Gg for HCl emissions in China)." These findings pertain to diverse years and encompass various source sectors. The methodology employed in this study is the "emission factor" method, aligning precisely with the approaches adopted in previous works (Fu et al., 2018; Hong et al., 2020; Yi et al., 2021; Yin et al., 2022); however, the references have not been properly cited in describing all the methods. Activity data were sourced from various references including statistical yearbooks, government statistics, and Gaode's POI data (Line 120); whereas emission data were curated and chosen from the existing literature (Line 121).

4. The manuscript refers to "41 specific source categories" concerning anthropogenic chlorine emissions. However, it remains unclear what precisely these 41 specific source categories encompass? Figure1 illustrate 7 primary categories and 24 distinct sources; Figure 3 demonstrate 5 economic sectors and 13 sources. In Table S1, there are 33 sub-categories noted. Upon examination of Table 2, only 24 source categories are evident. The classifications appear to be rather confusing. Additionally, why do the authors aggregate the emission source sectors from 7 to 5? If the authors divide the 7 main categories into 5 economic sectors. For restaurant sources, swimming pools, water treatment, and wastewater sources, to which sector do they belong to? These are missing in Figure 3.

5. The primary contribution to chlorine emissions in the outcomes of this study arises from biomass burning. However, instead of relying on FINN/GFED/GFAS data, this research utilized the "percentage of biomass domestic burning and open burning by province." This data from Table S6 are referenced from Zhou (2017). It's noteworthy that this information may not accurately reflect the conditions in the year 2018. Also, only using statistical data to estimate emissions of biomass burning and allocate the emissions spatially and temporally will raise large uncertainties.

6. Regarding the industrial production process, could you clarify how many industries are encompassed within this category? As it stands, it appears that only four specific types of industries are accounted for, namely cement, iron, steel and flat glass. However, there are additional industries, such as chemical industries, which are

known to release chlorine. How have these industries been addressed by the authors? Furthermore, from my perspective, "iron" and "steel" could arguably be regarded as a single industry. What's the difference here by separating them into two specific industries? Regrettably, the provided information lacks details.

7. Some of the key parameters employed in this study are quite old. For instance, in Line 175: "the value 2.2 g kg$^{-1}$ reported by Emmel et al. (1989)". "$\eta$d is the chlorine removal efficiency of dust removal facilities (25.1%), and $\eta$s is the chlorine removal efficiency of sulfate-removal facilities (95.5%)". In Lines 138-140: "we adopted the data from the study of Fu et al. (2018), which is the value of consumed coal considering the coal transportation… The values of X, R, $\eta$d, and $\eta$s can be found in Table 2 of our previous study (Liu et al., 2018). $\rho$ is the chlorine proportion of HCl (86.33%), fine particulate Cl- (10.09%), and Cl2 (3.58%) in emitted flue gases based on the local measurement (Deng et al., 2014)." Line 185: "$\eta$ is the removal efficiency of PM$_{2.5}$ (99%) in the garbage incineration station (Nan, 2016)" In Line 190-195, the authors adopted data from Fu et al (2018), while those data are for year 2014… It's worth noting that these datasets hold the potential for inducing overestimations and might not accurately reflect the circumstances of the year 2018.

8. "Water treatment" and "Tap water use", are there any double counting? Water treatment should include "tap water use".

9. Spatial allocation: Addressing emissions from other point sources where detailed information is unavailable, a uniform distribution across each individual point within each province has been employed. This approach might be perceived as somewhat arbitrary. Could you confirm whether emissions from industries are also apportioned in an averaged manner? If the primary sources stem from biomass burning, could you please elaborate how to conduct the allocation of emissions from biomass burning? It appears that the authors did not incorporate the geographical information of fire spots for open biomass burning. If population density was employed, it could potentially introduce significant uncertainties.

10. The discussions concerning temporal variations appear to be limited in depth. For instance, the substantial increase in HCl emissions in October compared to January, where the former is three times higher, and pCl emissions in October being five times that of January, lack sufficient justification. It's imperative to provide a reasoned explanation for these discrepancies.

11. Comparison with previous studies: the discussions provided lack specificity. For instance, the statement "The total HCl emission in this study is comparable with those estimated in the study of Fu et al. (2018) but with different contributions from source categories" requires more detailed clarification. "The HCl produced by coal combustion in this study is ~2 times higher than their estimation, which is mainly due to the different emission factors and control technology". Given that Fu's study

was conducted for year 2014 and this current study pertains to 2018, it's important to address the significant increase of HCl emissions, especially in light of the advancements in control technology over the intervening years. The contribution of emission factors and control technology to this twofold discrepancy needs to be explicitly outlined. The explanation for the lower HCl emissions from waste incineration being attributed to the use of more detailed and lower open-air combustion rates in various provinces could be elucidated further. Similarly, stating that higher HCl emissions from biomass burning are due to different estimation methods of household combustion rate and open combustion should be expanded upon for better clarity. When comparing your study's HCl emissions estimation with Zhang et al. (2022), discussing how your lower estimations were achieved due to factors such as coal combustion and waste incineration estimation is a good start. However, it would be beneficial to explicitly mention whether these adjustments have led to reduced uncertainties and whether your results can be considered more accurate compared to Zhang et al. (2022).

For $Cl_2$ emissions, explaining why you adopted relatively higher release ratios of chlorine for coal combustion and the factors contributing to the higher emissions from the usage of disinfectant is important.

Additionally, there are some papers reporting chlorine emission for different emission sectors for various regions, eg, Li et al (2020) for Shanghai; Yi et al (2021) for YRD; Qiu et al (2019) for Beijing. The results can also be compared and inserted to the table.

12. "Only this study and Yin et al. (2022) considered the emissions from cooking. The emission from cooking in this study is lower due to lower flue gas flux and shorter cooking durations". Qiu also considers cooking (Qiu, et al., Atmos. Chem. Phys. 2019, 19, 6737−6747).

Specific comments:

1. Lines 35-40: The term "chlorine atoms" is defined as "Cl," which could lead to confusion with the "Cl free radical" mentioned in lines 45-50. To avoid any ambiguity, please consider clarifying this terminology or employing an alternate term for either "Cl free radical" or "chlorine atoms."

2. Lines 53-54: The authors mentioned that "However, research on anthropogenic chlorine emission inventories in China is currently limited, and the temporal and spatial distribution of these emissions remains unclear". These descriptions are inaccurate. There are papers reporting temporal and spatial distribution of chlorine emissions (Fu et al., 2018; Yin et al., 2022).

3. Lines 54-57: The authors mentioned that "Consequently, anthropogenic chlorine emissions are rarely considered in numerical simulations of air quality, making it

challenging to study the chemical mechanism of chlorine and quantify the contribution of anthropogenic chlorine emissions to ozone and other pollutants using models". These descriptions are inaccurate. There are some papers that have already conducted the modeling study, but they are not properly cited (eg. Choi et al., Environ. Sci. Technol. 2020, 54, 13409−13418; Li et al., Environ. Sci. Technol. 2021, 55, 13625−13637; Wang et al., Atmos. Chem. Phys., 21, 13973–13996, 2021; Li et al., Journal of Geophysical Research: Atmospheres, 125, e2019JD032058. https://doi.org/10.1029/2019JD032058; Li et al., Journal of Geophysical Research: Atmospheres, 126, e2020JD034175. https://doi.org/10.1029/2020JD034175; Wang et al., Cite This: Environ. Sci. Technol. 2020, 54, 9908−9916; …).

4.  Lines 75-80, lack punctuation: "and waste incineration The study pointed out."

5.  Line 77, missing "." after "waste incineration".

6.  Line 80: It defines "particulate Cl-" as "(pCl)," which has already been defined in line 45. To avoid redundancy, please refrain from repeating this definition.

7.  Line 85: "chlorinecontaining" should be "chlorine-containing".

8.  Line 115: The provided emission ratios of 0.84 and 0.11 for disinfectant use sources HOCl and Cl2, respectively, require an explanation regarding the origin of this data.

9.  Line 120: To ensure clarity, it's essential to delineate the distinction between "activity data" and "emission data." Additionally, providing a precise definition for "emission data" would enhance understanding.

10. Lines 175-180, the numerical values for $\eta_d$ and $\eta_s$ need references.

11. Line 195 mentions that the proportion of open burning of solid waste varies by location. Are there specific values listed in the appendix or a reference for this information?

12. Lines 285-305: It appears that formulas (15) and (16) might be repetitive. Considering that the study encompasses both semi-standard and non-standard swimming pools, totaling 72%, could these categories be combined for the purpose of calculation? The study initially classifies swimming pools into public and private, and subsequently differentiates them as indoor and outdoor pools. Is it accurate to assume that the ratios of indoor and outdoor pools are evenly distributed between public and private ones? Additionally, does the value assigned to outdoor pool openings throughout the year seem excessively high?

13. Line 317: Considering the total health expenditure and the corresponding ratio, it is estimated to be 11,898.0 L in 2018. How is it derived?

14. 10. Line 325, formula (19) does not provide the chlorine disinfectant concentration for the aquaculture industry. What are the proportions of HOCl and $Cl_2$ in this case?

15. Line 340: It's possible that chlorine disinfectant use in household toilets is lower compared to public restrooms. Could the assumption of a 2 times higher chlorine disinfectant use in public toilets potentially be an overestimation? I recommend estimating the quantity of chlorine disinfectant utilized per household (e.g., per bottle of bleach) based on population, and subsequently comparing it against the emission estimate rooted in the 2 times higher value. This analysis can help identify any potential instances of overestimation.

16. Line 375, please provide the emission ratios for HOCl and $Cl_2$ during pesticide application.

17. Lines 442-448, the authors presented per-unit-area and per-capita emissions. However, it remains unclear what reasons contribute to these results? Within the discussion section, the paper predominantly showcases data results, yet falls short in delving into a comprehensive exploration of the underlying reasons.

18. Line 473-475, it is stated that "The emissions of HCl and fine particulate Cl exhibit relatively higher levels during early summer and autumn, coinciding with the frequent occurrence of biomass burning". However, it's important to note that the time period of biomass burning varies across different regions.

19. Line 544-545, "The inventory can be enhanced by including emissions from other anthropogenic activities that release chlorine. For example, the disposal and combustion of medical waste, which often contains high levels of plastic, can result in the release of significant amounts of active chlorine" isn't medical waste included in this study?

20. Line 554-555: "In this study, we developed a Chinese anthropogenic chlorine emissions inventory (ACEIC 2018) using emission factors mainly based on local measurements", this is inaccurate, as there are no measurements presented in this paper.

21. In section 3.3, it is suggested to provide reasons for higher chlorine emissions in different provinces to enhance the results analysis.

22. Line 740 mentions missing information on the meaning of the green line in Figure 4.

23. Line 760 suggests changing "Power" in the "Subsector" in Table 3 to "coal combustion."

24. Upon observing Figure S1, it raises the question of why the Per-unit-area emissions of Cl2 and HOCl are notably elevated in Shanghai.

---

## Author Comment (AC1)

This work compiled a comprehensive anthropogenic chlorine emission inventory for China. It improved the authors' previous ACEIC inventory by including the sources from cooking, usage of disinfectant, and pesticide. The paper provides valuable data for chlorine emissions in China. Some parts of the calculation and discussion are not very clear and need improvement.

Response: We express our gratitude to the referee for offering a reflective and comprehensive evaluation of our paper. The referee's suggestions have significantly contributed to the enhancement of this manuscript. Below, we present point-by-point responses to the referee's remarks and summarize the modifications that have been implemented in the revised manuscript.

Major comments:

[Comment]: 1. I would suggest to rearrange the introduction to make it clearer. Paragraph 2-5 have some overlaps and inconsistent, making the logic confusing.

Response: Thanks for your valuable suggestion. We agreed and have rearranged the introduction to provide a clearer presentation. Please see Section 1 (Introduction) in the revised manuscript.

[Comment]: 2. The most significant finding of this work that differs from previous studies may be a large source of HOCl and $Cl_2$ from the usage of chlorine-containing disinfectants. It seems that the estimate of this part of emissions (section 2.2.6) assumes that chlorine gases volatilized from the water will be directly released to the atmosphere. However, many water treatment plants, hospitals, and swimming pools are indoors, some of the waste gases are also treated. This needs more discussion.

Response: Thanks for pointing out this important issue. Currently, there are no specific requirements or applicable standards for the management of waste gas from disinfection

in China. Consequently, a majority of waste gases are released without systematic regulation. $Cl_2$ and HOCl released from the usages of chlorine-containing disinfectants are known for their pungent odors, which can pose risks to human health. As a result, indoor disinfection procedures, such as those in hospitals and indoor swimming pools, require meticulous attention to ventilation (Huang, 2012; Tang, 2003; the standard GB 15982-2012). During the ventilation process, there is a rapid exchange of indoor and outdoor air, resulting in the release of chlorine gases into the atmosphere. We have added this discussion in Section 2.2.6 as follows: "$Cl_2$ and HOCl released from the usages of chlorine-containing disinfectant are known for their pungent odors, which can pose risks to human health. Consequently, indoor disinfection generally requires meticulous attention to ventilation, such as in hospitals and indoor swimming pools (Huang, 2012; Tang, 2003; the standard GB 15982-2012)".

Reference:

Huang, Y.: Study of the Natural Ventilation Strategy of Hospital Clinic Waiting in Lingnan Regions, M.S. thesis, South China University of Technology, China, 125 pp., 2012.

Tang, J.: Design of the Air-Conditioner for Chamber Indoor Swimming Pool, Mechanical and Electrical Equipment, 5, 17-20, 2003 (in Chinese).

[Comment]: 3. For the part of comparing with other works, the authors frequently attributed the difference to the use of different methods without other explanation. Please provide a clearer discussion. For instance, why different methods are used? which one is better? suggestions to reduce the uncertainties of the methodology ....

Response: Thanks for your valuable suggestion. We agreed and have made a major revision for discussing the comparison with other studies. Please see Section 4.1 in the revised manuscript.

[Comment]: 4. Please make sure all the numbers used in the calculations have proper references.

Response: Your suggestion is greatly appreciated. We have included proper references for all the numbers used for emission calculations throughout the revised manuscript.

Specific comments:

[Comment]: 5. For the name RCEI and statement such as in line 46, it is hard to say whether HCl and pCl can be grouped into reactive chlorine species as they are not that reactive and fast producing Cl radicals in the atmosphere.

Response: Thanks for your valuable suggestion. We agree with the perspective put forth by the reviewer. Consequently, we have substituted "reactive chlorine species" with "chlorine species" throughout the revised manuscript.

[Comment]: 6. Paragraph 2, you said research on anthropogenic chlorine emission in China is very limited and rarely considered in air quality simulations, but later you provide a series of examples in paragraph 3 and 4. I would suggest removing those statements and merging them with paragraph 4.

Response: Thanks for your valuable suggestion. To provide a logical presentation, we have removed these statements in paragraph 2, carried out the necessary modifications and restructured the introduction section. Please see Section 1 in the revised manuscript.

[Comment]: 7. Paragraph 3, this part starts with saying emission inventory in foreign countries, but no related information are introduced.

Response: Thanks for pointing out this issue. To avoid confusing, we have revised this sentence in Section 1 as follows: "The study of estimating the anthropogenic chlorine emission in China began by Mcculloch et al. (1999), who established a global

anthropogenic chlorine emission inventory called Reactive Chlorine Emissions Inventory (RCEI) based on a relatively rough statistical dataset in 1990".

[Comment]: 8. line 87, which sources were overlooked, such as?

Response: Thanks for your question. The overlooked sources in previous studies include environmental disinfection, tap water utilization, pesticide application, etc. We have added this information in Section 1 as follows: "Such overlooked emissions include those from environmental disinfection, tap water utilization, and pesticide application, and etc.".

[Comment]: 9. line 89, you may want to summarize the pros and cons of these studies at least briefly before this statement. Why did the previous estimates differ so largely? What are the uncertainties?

Response: We appreciate your valuable suggestions. We have provided a specific discussion on the previous studies in Section 1 as follows: "Despite these studies, the anthropogenic chlorine emission in China remains uncertain and further investigation is still warranted. Firstly, the estimated chlorine emissions in China varied in different studies due to the different applications of emission factors and estimation methods. Some studies have utilized emission factors derived from foreign sources or standards and guidelines that may not accurately reflect the specific local conditions in China. Some calculation methods are rudimentary and lack the granularity needed to effectively capture variations among provinces or different sources. Secondly, some modeling studies (Choi et al., 2020; Li et al., 2021) have used the anthropogenic chlorine emission as inputs and found that the simulated concentrations of chlorine species (HCl and pCl) were underestimated against the observation, which suggests that there are large uncertainties or missing sources for the current emission estimation. Lastly, chlorine emissions from anthropogenic activities were reported in the recent literature, but they have not been considered in the developed emission inventory for

China. Such overlooked emissions include those from environmental disinfection, tap water utilization, pesticide application, and etc. The neglect of these sources can lead to an underestimation of the total chlorine emission. As a result, the development of a comprehensive anthropogenic chlorine emission inventory that addresses the above issues is of great significance in reducing the uncertainty of the emission estimation".

[Comment]: 10. line 92, what is basic data? Do you mean activity data?

Response: Thanks for this valuable question. Here the basic data means the activity data. To clarify the description, we have replaced "basic data" with "activity data" in Section 1.

[Comment]: 11. line 116, please provide references for the release ratio.

Response: Thank you for your valuable suggestion. In the revised manuscript, we mentioned the value of release ratios for usages of chlorine-containing disinfectants and pesticides with references in Section 2.2.6 and 2.2.7, respectively. Here, we removed this introduction in Section 2.1.

[Comment]: 12. line 144, based on your statement, MM should equal to 0.5 for $Cl_2$ as you defined it as the ratios of the molar mass of chlorine atom to the molecular weight of chlorine species.

Response: Thanks for pointing out this issue. The term "MM" is the chlorine content in the chlorine species. As a result, it is 35.5/36:5, 1, 1 for HCl, fine particulate Cl-, and $Cl_2$, respectively. We have revised the text in Section 2.2.1 as follows: "MM refers to the chlorine content in chlorine species (35.5/36:5, 1, 1 for HCl, pCl, and $Cl_2$)".

[Comment]: 13. line 177-178, please provide references for the chlorine removal efficiencies.

Response: Thank you for your suggestion. We have provided the reference (Liu et al., 2018) for the chlorine removal efficiencies in the revised manuscript in Section 2.2.3.

Reference:

Liu, Y. M., Fan, Q., Chen, X. Y., Zhao, J., Ling, Z. H., Hong, Y. Y., Li, W. B., Chen, X. L., Wang, M. J., and Wei, X. L.: Modeling the impact of chlorine emissions from coal combustion and prescribed waste incineration on tropospheric ozone formation in China, Atmos Chem Phys, 18, 2709-2724, 10.5194/acp-18-2709-2018, 2018.

[Comment]: 14. Section 2.2.6, the whole part involves assumptions that the water in the facilities (water treatment plant, hospital, swimming pool, etc.) are open to the atmosphere and the waste gases are released without any treatment. This doesn't sound very true.

Response: Thanks for your valuable comment. Please see our response to the Comment 2.

[Comment]: 15. Section 2.2.6b, it is not clear how the calculation was conducted.

Response: Thank you for your suggestion. The emissions are estimated using Formula 12. We have specified the calculation method in Section 2.2.6b as follows: "The emission of water treatment is estimated using formula 12. The water amount that needs disinfection ($W_j$) is the quantity of tap water supplied in each province, which can be obtained from the China Urban and Rural Construction Statistical Yearbook 2019 (National Bureau of Statistics, 2019a). The emission factors can be found in Table S5."

[Comment]: 16. line 496, why are the emission factors and control technologies different? Are your estimations better? Same for the entire section 4.1, when you said different results were due to different calculation methods, could you please elaborate more and maybe demonstrate that your methods are more appropriate?

Response: Thank you for your valuable comment. Regarding the emission factors for coal combustion, this study relied on the local measured and survey data from the literature. Compared with Fu et al. (2018), who relied on control technology application ratios assumed from national policies, Yin et al. (2022), who used foreign application ratios, and Zhang et al. (2022), who employed an overestimation of application ratios using an S-curve formula, this study's selection of control technology application ratios is based on the domestic research literature, rendering it a more reasonable choice. For open waste incineration, this study estimates the open burning ratio for each province based on the proportion of open waste burning from the statistical yearbooks. Relative to Yin et al. (2022), this study offers a more refined differentiation for each province, aligning it more closely with real conditions. In the case of $PM_{2.5}$ emission factors for biomass combustion, Fu et al. (2018) and Yin et al. (2022) relied on data from the Guidelines for Compilation of Atmospheric Pollutant Emission Inventories for Biomass Combustion, while this study referenced literature from field observations, thereby making the estimations more reasonable. For restaurant emissions, this study adheres to lower flue gas flow rates, shorter cooking durations, and a lower $Cl^-$ proportion in $PM_{2.5}$, which was obtained through local measurements and is in accordance with national standards and actual circumstances. Consequently, this approach reduces cooking emissions compared with Qiu et al. (2019) and Yin et al. (2022). In the case of swimming pools, while Yin et al. (2022) used data from the standard, this study drew upon data from the experimental research literature. Furthermore, this study has provided a more detailed breakdown of indoor and outdoor swimming pool operating hours and dosages. We have made specific modifications and explanations in Section 4.1.

[Comment]: 17. Figure 2 provides the same information as Figure 1d. Why put it as an individual figure? Also, it looks like the proportion of chlorine emissions from different sources of disinfection, not the proportion of actives as the figure title described.

Response: Thanks for your careful review. Figures 1d and 2 are two different figures. Figure 1d is to show the contribution of different source categories to the total HOCl emission, including emissions from usages of chlorine-containing disinfectants and pesticide. Figure 2 is to show the contribution of different usages of chlorine-containing disinfectant to the total emission of disinfection process, in which the emission from pesticide usages is excluded. To avoid confusion, we have revised the caption of Figure 2 as follows: "Figure 2 Proportion of chlorine emissions from different usages of chlorine-containing disinfectant in China".

[Comment]: 18. Figure 4: please introduce the green line in the label or figure captain.

Response: Thanks for this comment. The green line in Figure 4 represents the emissions for each province. We have included this information in the caption of Figure 4 as follows: "Figure 4 Emissions (green line) and contribution proportions of HCl (a), fine particulate $Cl^-$ (b), $Cl_2$ (c), and HOCl (d) by province in 2018."

[Comment]: 19. Figure 5: it is quite strange to use the unit of Mg/grid/yr, especially when no grid information is provided in figure captain. I would suggest using a unit of Mg/m2/yr or something similar.

Response: Thanks for this valuable comment. We have incorporated information about the grid resolution ($0.1° \times 0.1°$) into the caption of Figure 5 as follows: "Figure 5 Spatial distribution of anthropogenic HCl, fine particulate $Cl^-$, $Cl_2$, and HOCl emissions in 2018 at a $0.1° \times 0.1°$ resolution."

[Comment]: 20. Table 6: could you please also include emission numbers or ranges? I am not clear what useful information can be provided by comparing uncertainties with different studies.

Response: Thank you for your suggestion, which we agree. Instead of Table 6, we drew Figure 7 which included the emission and its range of uncertainty for different emission inventories. Generally, the percentage of uncertainty has slightly reduced in this study. We have revised the text in Section 4.2 as follow: "The uncertainties for HCl, pCl, Cl$_2$, and HOCl emissions were estimated at a 95% confidence interval, resulting in percentage ranges of -48% to 45%, -59% to 89%, -44% to 58%, and -44% to 79%, respectively (Figure 7). It can be seen that the estimated emissions of HCl and pCl are within the uncertainty ranges of other studies. Due to the additional sources of Cl$_2$ and HOCl in this study, the emissions are relatively higher compared with Yin et al. (2022). However, the percentage of uncertainty for all chlorine species generally reduces in this study compared with the other studies."

[Figure]

Figure7 Comparison of anthropogenic chlorine emissions and uncertainty ranges (blue text) with other studies.

---

## Author Comment (AC2)

Review comments on the manuscript titled "ACEIC: a comprehensive anthropogenic chlorine emission inventory for China"

The chlorine radical assumes a pivotal role in atmospheric chemistry, exerting a significant influence on atmospheric oxidation capacity, thereby contributing to secondary air pollution. Chlorine emanates from diverse sources, encompassing both anthropogenic and sea salt aerosol. While preceding studies have partially addressed certain aspects of chlorine emissions, a more comprehensive and reliable dataset consisting complete chlorine species and their respective source contributions remains a pressing necessity. The present study developed an emission inventory detailing anthropogenic chlorine sources, including HCl, $Cl_2$, pCl and HOCl, across 7 distinct source sectors in China for the year 2018 by using the emission factor methodology. Although the topic is both interesting and significant, the manuscript appears to be hindered by a lack of novelty. Several existing papers have already reported chlorine emissions, spanning cities, regions and even the entire nation of China, encompassing a multitude of sources. The current study closely aligns with these prior works, employing nearly identical methods, activity data and reported emissions factors. Consequently, the findings and uncertainties presented within this study are largely consistent with those from earlier investigations. Therefore, the identification of genuinely novel insights or contributions to the scientific community proves to be a challenge.

Response: We thank the referee for providing a thoughtful and detailed review of our paper. We have carefully considered the valuable suggestions and major revisions of this study are shown as follows: 1) The abstract and introduction sections have been reorganized to highlight the novelty of this study. Compared with previous studies, this updated inventory considered more anthropogenic sources, used more localized emission factors, and adopted more refined estimation method. 2) We improve the emission inventory by using the fire data from the Himawari-8 satellite to allocate the provincial open biomass burning emissions spatially and temporally. 3) We provide a deeper discussion on the comparison of this updated inventory to the previous studies. The advantage of this inventory is highlighted. The referee's comments have helped to

improve this manuscript. Below, we provide a point-by-point response to the referee's comments and summarize the changes that have been made in the revised manuscript.

Major concerns:

[Comment]: 1.The primary objective of this study is to establish an extensive inventory encompassing anthropogenic chlorine emissions in China. This inventory includes HCl, fine particle Cl$^-$, Cl$_2$, and HOCl from 7 anthropogenic sources for year 2018. Nonetheless, the clarity in communicating the novelty and distinctiveness of this study from its precursors is lacking. Several papers have already reported on China's chlorine emissions inventory for various years, such as 2014 (Fu et al., 2018), 2019 (Yin et al., 2022), and even the range spanning 1960 to 2014 (Zhang et al., 2022). Particularly, the work by Yin (2022) has extensively detailed emissions of HCl, pCl, Cl$_2$, and HOCl, with source categories covering 22 sectors. Notably, they employed a spatial resolution of 0.1° × 0.1° and reported the temporal resolution as well. The outcomes of the present study remain consistent with those presented in previous investigations (Fu et al., 2018; Zhang et al., 2022; Yin et al., 2022), and the level of uncertainties closely mirrors that of earlier researches, as indicated in Table 6. It is pertinent to highlight that this study adopts a methodology akin to prior works, employing the emission factor method, which does raise concerns about potential duplication of prior research efforts.

Response: Thank you for this question. However, this study is not a repetitive work. Compared with previous studies, we have made innovations and improvements in the following three aspects:

1. Expanded emission sources: This study has augmented the range of emission sources. For instance, it has included Cl$_2$ emissions from flat glass production, emissions from urban open waste incineration, emissions from municipal tap water usage and wastewater treatment in county towns, emissions from tap water usage during disinfectant application, emissions during environmental disinfection processes, and emissions of chlorine-containing substances from pesticide use.

2. Localized emission factors: Certain emission factors have been localized in this

study. For example, for PM$_{2.5}$ emissions from biomass combustion, while Yin et al. (2022) and Fu et al. (2018) relied on emission factors from the Guidelines for Compilation of Atmospheric Pollutant Emission Inventories for Biomass Combustion, this study primarily drew upon local measured data from the literature. This localization makes the emission factors more aligned with real-world conditions, thus reducing uncertainty. For the emission of cooking, the study adopted lower flue gas flow rates, shorter cooking durations, and a lower Cl$^-$ proportion in PM$_{2.5}$, based on national standards and actual circumstances, which results in reduced cooking emissions compared with Qiu et al. (2019) and Yin et al. (2022). Additionally, for chlorine dosing in swimming pools, this study referenced experimental research literature, whereas Yin et al. (2022) relied on national standards.

3. Refined estimation method: The estimation methodology has been refined, rendering the results more detailed and realistic. For instance, in the case of waste incineration, different provinces are assigned different open burning rates, enhancing the accuracy of the inventory for open waste incineration. For swimming pool emissions, the study distinguishes the opening days between indoor and outdoor pools, in contrast to Yin et al. (2022), who assumed all pools only open in summer, making this study more realistic and detailed.

Hence, this study does not duplicate previous efforts but rather innovates and improves upon prior research. It expands emission sources, localizes certain emission factors, and refines calculation methods, reducing errors and uncertainties in the inventory, thereby enhancing the completeness of the anthropogenic chlorine emissions inventory for China. Please see the revision in Section 1 and Section 4.

[Comment]: 2. The authors have highlighted that there exists uncertainty regarding previous anthropogenic sources, necessitating further investigation. However, the specific nature of these uncertainties and the areas requiring further inquiry remain unclear. Has the current research succeeded in mitigating any of these uncertainties? If so, through what means has this reduction been achieved? Have the authors

incorporated more precise activity data, or have they embraced empirically measured emission factor data? It appears that the authors predominantly relied on activity data from statistical yearbooks and incorporated emissions factor information gleaned from previous literature. Consequently, the uncertainties associated with HCl, pCl, $Cl_2$, and HOCl emissions spanned a range of -48% to 45%, -59% to 89%, -44% to 58%, and -44% to 79% respectively. These figures closely mirror those from earlier studies, as also evidenced in Table 6.

Response: Thank you for this question. To ameliorate the uncertainty associated with this inventory, our study has undertaken the following concerted efforts.

Firstly, with respect to emission sources, our research surpasses the scope of Yin et al. (2022) by considering $Cl_2$ emissions from flat glass production, emissions from urban open burning of waste, emissions resulting from municipal tap water usage and wastewater treatment in counties, emissions during the use of disinfectants that include tap water, and emissions during environmental disinfection and pesticide use. These additions rectify underestimations and reduce a certain degree of uncertainty stemming from insufficient consideration of sources.

Secondly, we have localized improvements in certain emission factors, thereby mitigating uncertainties. While Yin et al. (2022) and Fu et al. (2018) relied on $PM_{2.5}$ emission factors for biomass burning derived from the Guidelines for Compilation of Air Pollutant Emission Inventories for Biomass Burning, our study predominantly relies on emission factors sourced from field measurements, rendering them more congruent with actual conditions and reducing uncertainty. For the cooking emission, we have adopted lower smoke flow rates and shorter cooking durations in accordance with national standards and practical circumstances. Additionally, we have employed $Cl^-$ in $PM_{2.5}$ ratios lower than those derived from local measurements compared with Qiu et al. (2019), thereby reducing cooking emissions. Moreover, regarding the emission factor of the swimming pool, while Yin et al. (2022) based their calculations on national standards, our study draws from experimental research literature, aligning more closely with actual chlorine addition practices.

Lastly, we have refined the estimation methodology to yield more detailed and

reasonable results, thus diminishing uncertainty. In the context of waste incineration, we have differentiated between provinces with varying rates of open incineration, augmenting the completeness and precision of the inventory for this aspect. Similarly, with respect to swimming pool emissions, we have considered the varying opening days of indoor and outdoor pools, in contrast to Yin et al. (2022) who assumed that all pools only open during the summer season.

In summary, the innovations in our study primarily aim to reduce errors and uncertainties in the inventory, thereby enhancing the comprehensiveness and meticulousness of the anthropogenic chlorine emission inventory. We have revised the text in Section 4.2.

[Comment]: 3. The authors emphasized that "it should be noted that some important sources of chlorine emissions have been overlooked, leading to large uncertainties in recent decade estimates (219-707 Gg for HCl emissions in China)." These findings pertain to diverse years and encompass various source sectors. The methodology employed in this study is the "emission factor" method, aligning precisely with the approaches adopted in previous works (Fu et al., 2018; Hong et al., 2020; Yi et al., 2021; Yin et al., 2022); however, the references have not been properly cited in describing all the methods. Activity data were sourced from various references including statistical yearbooks, government statistics, and Gaode's POI data (Line 120); whereas emission data were curated and chosen from the existing literature (Line 121).

Response: Thank you for this question. Firstly, all the activity data and emission factors and their sources can be found in the supplementary information. We have also added citations for the critical activity data and emission factors in the revised manuscript. We have revised the text in Section 2.2 as follows: "This emission inventory uses a large amount of activity data and advanced emission factors. Most of the activity data, emission factors, and related references can be found in Tables S3-S12. Generally, the activity data were obtained from the yearbook (e.g., China Energy Statistical Yearbook, China Industry Statistical Yearbook, and China Urban-Rural Construction Statistical

Yearbook), government statistics (e.g., National Bureau of Statistics, and General Administration of Sport of China), and Gaode's POI data. The emission factors were mainly based on the measured and survey data from the literature".

Secondly, even though we employed the same "emission factor" methodology, there are variations in the selection of emission factors and specific estimation methods in different studies. Some of them have been improved in this study compared with previous studies. For example, the estimation of the combustion ratio for the open waste burning and the estimation of opening hours for the swimming pool have seen innovative improvements in this study. Please see our response to Comment 1 and 2.

[Comment]: 4. The manuscript refers to "41 specific source categories" concerning anthropogenic chlorine emissions. However, it remains unclear what precisely these 41 specific source categories encompass? Figure1 illustrate 7 primary categories and 24 distinct sources; Figure 3 demonstrate 5 economic sectors and 13 sources. In Table S1, there are 33 sub-categories noted. Upon examination of Table 2, only 24 source categories are evident. The classifications appear to be rather confusing. Additionally, why do the authors aggregate the emission source sectors from 7 to 5? If the authors divide the 7 main categories into 5 economic sectors. For restaurant sources, swimming pools, water treatment, and wastewater sources, to which sector do they belong to? These are missing in Figure 3.

Response: Thank you for pointing out this issue. In the revised supplementary information, Tables S1 and S2 show the aggregation of 41 specific sources into 7 primary source categories and 5 economic sectors, respectively. We have added this information in Section 2.1 as follows: "In our study, we have compiled the emissions of chlorine species (HCl, pCl, $Cl_2$, HOCl) from 41 anthropogenic activities (Table S1 and S2) across the 31 provinces in mainland China. These emissions are categorized into seven major source categories (Table S1)", Section 3.2 as follows: "We aggregated the anthropogenic chlorine emissions from 41 specific sources into 5 economic sectors, including power, industry, residential, agricultural, and biomass burning (Table S2)",

the caption of Figure 1 as follows: "The aggregation of 7 primary source categories from 41 specific sources can be found in Table S1", and the caption of Figure 3 as follows: "The aggregation of 5 economic sectors from 41 specific sources can be found in Table S2". Previous studies (e.g., Yin et al., 2022) on chlorine emissions have classified specific sources into 7 primary source categories based on the energy consumption. In this study, besides this classification method, we aggregated the specific sources into economic sectors. This categorization is also adopted for the widely-used emission inventory in China (e.g., MEIC) and it can attract the interest of potential readers of government officials and economic experts. According to Table S2, restaurant sources, swimming pools, water treatment, and wastewater sources belong to the residential sector.

Reference:

Yin, S., Yi, X., Li, L., Huang, L., Ooi, M. C. G., Wang, Y., Allen, D. T., and Streets, D. G.: An Updated Anthropogenic Emission Inventory of Reactive Chlorine Precursors in China, ACS Earth and Space Chemistry, 6, 1846-1857, 10.1021/acsearthspacechem.2c00096, 2022.

[Comment]: 5. The primary contribution to chlorine emissions in the outcomes of this study arises from biomass burning. However, instead of relying on FINN/GFED/GFAS data, this research utilized the "percentage of biomass domestic burning and open burning by province." The data from Table S6 are referenced from Zhou (2017). It's noteworthy that this information may not accurately reflect the conditions in the year 2018. Also, only using statistical data to estimate emissions of biomass burning and allocate the emissions spatially and temporally will raise large uncertainties.

Response: Thank you for this question. The activity data for biomass burning emission estimation can be derived from statistical data or satellite data. Both methods have their advantages and disadvantages. Compared with the method using satellite data, the method based on statistical data can estimate the household burning, and avoid missing the small or short-term fire incidents that cannot be detected by the satellite. It's worth

noting that in recent years, many studies have employed this calculation method (Zhang et al., 2019; Li et al., 2019; Zhou et al., 2019; Wang et al., 2018; Yan et al., 2006), which underscores the advantages of this approach. For the spatial and temporal allocation, in the revised manuscript, we used the fire location and fire radiation power over cropland from the Himawari-8 satellite data in 2018, instead of rural population density and empirical statistic data, to allocate the provincial open biomass burning emission to reduce the uncertainty. The spatial distribution maps for chlorine emission have been revised in the study.

We have added this description in Section 2.3 as follows: "For the emission of biomass open burning, we allocated the provincial emissions spatially to the fire location according to its fire radiation power over the cropland. The fire location and its fire radiation power data were derived from the Himawari-8 satellite data (https://www.eorc.jaxa.jp/ptree/userguide.html, last access: 1 January 2023)", and Section 2.4 as follows: "Based on the fire location and its fire radiation power over the cropland from the Himawari-8 satellite data, we performed temporal allocation of chlorine emissions from biomass burning for each province".

Reference:

Li, L., Zhao, Q., Zhang, J., Li, H., Liu, Q., Li, C., Chen, F., Qiao, Y., Han, J.: Bottom-up emission inventories of multiple air pollutants from open straw burning: A case study of Jiangsu province, Eastern China, Atmospheric Pollution Research, 10, 501–507, 2019.

Wang, J., Xi, F., Liu, Z., Bing, L., Alsaedi, A., Hayat, T., Ahmad, B., Guan, D.: The spatiotemporal features of greenhouse gases emissions from biomass burning in China from 2000 to 2012, Journal of Cleaner Production, 181, 801-808, 2018.

Yan, X., Ohara, T. Akimoto, H.: Bottom-up estimate of biomass burning in mainland China, Atmospheric Environment, 40, 5262-5273, 2006.

Zhang, X., Lu, Y., Wang, Q., Qian, X.: A high-resolution inventory of air pollutant emissions from crop residue burning in China, Atmospheric Environment, 213, 207–21, 2019.

Zhou, Z., Tan, Q., Deng, Y., Wu, K., Yang, X., Zhou, X.: Emission inventory of

anthropogenic air pollutant sources and characteristics of VOCs species in Sichuan Province, China, Journal of Atmospheric Chemistry, 76, 21–58, 2019.

[Comment]: 6. Regarding the industrial production process, could you clarify how many industries are encompassed within this category? As it stands, it appears that only four specific types of industries are accounted for, namely cement, iron, steel and flat glass. However, there are additional industries, such as chemical industries, which are known to release chlorine. How have these industries been addressed by the authors? Furthermore, from my perspective, "iron" and "steel" could arguably be regarded as a single industry. What's the difference here by separating them into two specific industries? Regrettably, the provided information lacks details.

Response: Thank you for this question. In Section 2.2.2, we have mentioned that five specific types of industries have been taken into account in this inventory: cement, iron, steel, flat glass, and HCl production. It's worth noting that HCl production belongs to the chemical industries. In the future, we are committed to conducting further research to incorporate additional sources of chlorine emissions from various chemical industrial processes into the inventory.

Regarding the separation of iron and steel, we have maintained this distinction due to the differences in their emission factors (please see Table S6), as outlined in the study of Yi et al. (2020). This separation allows us to provide a more accurate and detailed estimation of chlorine emissions from these two distinct processes.

Reference:

Yi, X., Yin, S., Tan, X., Huang, L., Wang, Y., Chen, Y., and Li, L.: Preliminary study on the inventory of sources of hydrogen chloride and particulate chlorine in the atmosphere in Shanghai, Acta Scientiae Circumstantiae (in Chinese), 40, 469-478, 10.13671/j.hjkxxb.2019.0376, 2020.

[Comment]: 7. Some of the key parameters employed in this study are quite old. For

instance, in Line 175: "the value 2.2 g kg$^{-1}$ reported by Emmel et al. (1989)". "$\eta d$ is the chlorine removal efficiency of dust removal facilities (25.1%), and $\eta s$ is the chlorine removal efficiency of sulfate-removal facilities (95.5%)". In Lines 138-140: "we adopted the data from the study of Fu et al. (2018), which is the value of consumed coal considering the coal transportation… The values of X, R, $\eta d$, and $\eta s$ can be found in Table 2 of our previous study (Liu et al., 2018). $\rho$ is the chlorine proportion of HCl (86.33%), fine particulate Cl$^-$ (10.09%), and Cl$_2$ (3.58%) in emitted flue gases based on the local measurement (Deng et al., 2014)." Line 185: "$\eta$ is the removal efficiency of PM$_{2.5}$ (99%) in the garbage incineration station (Nan, 2016)" In Line 190-195, the authors adopted data from Fu et al (2018), while those data are for year 2014… It's worth noting that these datasets hold the potential for inducing overestimations and might not accurately reflect the circumstances of the year 2018.

Response: Thank you for this question. The emission factors used in this study were derived from the relevant literature, with a preference for more recent and locally measured data whenever possible. However, it's acknowledged that due to the extensive scope of the work, some of the data might not be the most up-to-date, which can introduce uncertainties into the inventory. We are committed to continuously improving the emissions inventory. If more suitable data become available in the future, we will certainly update the inventory accordingly.

[Comment]: 8. "Water treatment" and "Tap water use", are there any double counting? Water treatment should include "tap water use".

Response: Thank you for this question. Water treatment and the usage of tap water are two different processes for chlorine emission. After undergoing water treatment, residual chlorine remains in the water, which is released during the process of using tap water. Hence, water treatment does not include the use of tap water, and there is no double counting.

[Comment]: 9. Spatial allocation: Addressing emissions from other point sources where detailed information is unavailable, a uniform distribution across each individual point within each province has been employed. This approach might be perceived as somewhat arbitrary. Could you confirm whether emissions from industries are also apportioned in an averaged manner? If the primary sources stem from biomass burning, could you please elaborate how to conduct the allocation of emissions from biomass burning? It appears that the authors did not incorporate the geographical information of fire spots for open biomass burning. If population density was employed, it could potentially introduce significant uncertainties.

Response: Thank you for your question. In the revised manuscript, we used the fire location and fire radiation power over cropland from the Himawari-8 satellite data in 2018, instead of rural population density and empirical statistic data, to allocate the provincial open biomass burning emission spatially and temporally. Such an allocation method can help reduce the uncertainty. Please see our response to Comment 5. Emissions from household biomass burning are still allocated based on rural population density.

For emissions from other point sources with unavailable installed capacity data, such as industrial production, we assumed a uniform emission for them and spatially allocated the provincial emissions evenly to each point. We acknowledge that this method may induce uncertainties. If we can obtain the production data from factories in the future, we will use this data as weighting factors for allocation.

[Comment]: 10. The discussions concerning temporal variations appear to be limited in depth. For instance, the substantial increase in HCl emissions in October compared to January, where the former is three times higher, and pCl emissions in October being five times that of January, lack sufficient justification. It's imperative to provide a reasoned explanation for these discrepancies.

Response: Thank you for your question. The temporal variation of HCl and pCl emission is mainly contributed by the biomass burning emission. In the revised

manuscript, we used the fire location and fire radiation power over cropland from the Himawari-8 satellite data to allocate the temporal variation of chlorine emissions from biomass open burning in various provinces. This modification can greatly reduce the uncertainty of the estimated temporal variation of the emission.

We have revised the text in Section 3.5 as follows: "Figure 6 shows the temporal variation of anthropogenic emissions for different chlorine species. For HCl and pCl, the emission in mainland China presents a bimodal variation. A remarkable peak is in early spring (February to April), and a small peak is in early autumn (August to October). The high emission in these months is attributed to the biomass burning emission with active agricultural activities. In contrast, emissions from other sectors remain relatively stable throughout the year. It's worth noting that the monthly variations vary across different regions because of the varied period of biomass burning, as shown in Fig. S5 and S6. For example, in Northeast China (Liaoning, Jilin, and Heilongjiang), where extensive straw burning occurs before crop planting, emissions are elevated in spring only."

[Comment]: 11. Comparison with previous studies: the discussions provided lack specificity. For instance, the statement "The total HCl emission in this study is comparable with those estimated in the study of Fu et al. (2018) but with different contributions from source categories" requires more detailed clarification. "The HCl produced by coal combustion in this study is ~2 times higher than their estimation, which is mainly due to the different emission factors and control technology". Given that Fu's study was conducted for year 2014 and this current study pertains to 2018, it's important to address the significant increase of HCl emissions, especially in light of the advancements in control technology over the intervening years. The contribution of emission factors and control technology to this twofold discrepancy needs to be explicitly outlined. The explanation for the lower HCl emissions from waste incineration being attributed to the use of more detailed and lower open-air combustion rates in various provinces could be elucidated further. Similarly, stating that higher HCl

emissions from biomass burning are due to different estimation methods of household combustion rate and open combustion should be expanded upon for better clarity. When comparing your study's HCl emissions estimation with Zhang et al. (2022), discussing how your lower estimations were achieved due to factors such as coal combustion and waste incineration estimation is a good start. However, it would be beneficial to explicitly mention whether these adjustments have led to reduced uncertainties and whether your results can be considered more accurate compared to Zhang et al. (2022). For $Cl_2$ emissions, explaining why you adopted relatively higher release ratios of chlorine for coal combustion and the factors contributing to the higher emissions from the usage of disinfectant is important.

Additionally, there are some papers reporting chlorine emission for different emission sectors for various regions, eg, Li et al (2020) for Shanghai; Yi et al (2021)for YRD; Qiu et al (2019) for Beijing. The results can also be compared and inserted to the table.

Response: Thank you for this question. The differences in HCl emissions from coal combustion in this study compared to previous research mainly stem from variations in emission factors and control technologies. Our study relies on emission factors sourced from field observations, in contrast to Fu et al. (2018), who utilized control technology application ratios based on national policies, Yin et al. (2022), who adopted foreign application ratios, and Zhang et al. (2022), who employed an overestimated application ratio derived from an S-curve formula. The control technology application ratios used in our study are based on domestic research literature, providing a more reasonable basis. Regarding open waste incineration, we estimated the open burning ratio for each province as 1 minus the treatment rate (sourced from the statistical yearbook), in contrast to Yin et al. (2022), who assumed a uniform open burning ratio of 0.05 for all provinces nationwide. Our approach allows for a more nuanced consideration of each province's unique circumstances and is better aligned with reality. In our study, higher HCl emissions from biomass combustion are primarily attributed to differing estimations of household combustion rates and outdoor combustion methods. Our household combustion rate is substantiated by literature with documented data sources. Additionally, our approach to open burning employs a bottom-up emission factor

method, which avoids underestimation resulting from the omission of small-scale or short-term fire incidents compared to satellite-based detection methods. Concerning the $PM_{2.5}$ emission factor for biomass combustion, Fu et al. (2018) and Yin et al. (2022) used data from the Biomass Burning Emissions Inventory Guidelines. In contrast, our study references literature with field observations, resulting in a more realistic estimation. For the restaurant sector, we adopted lower flue gas flow rates, shorter cooking times, and lower $Cl^-$ proportions in $PM_{2.5}$, compared to Qiu et al. (2019), which led to reduced cooking emissions.

Regarding $Cl_2$, our chlorine release rate is based on a specific study (Deng et al., 2014) and derives from on-site experiments, lending it greater credibility. Both Fu et al. (2018) and Yin et al. (2022) referenced this article, but their chlorine release rates were incorrect, resulting in duplicated calculations. For swimming pools, Yin et al. (2022) used standard data, while our study incorporated data from the experimental research literature, further delineating the indoor and outdoor pool opening times and dosages. This addition resulted in higher disinfectant emissions than Yin et al. (2022). Furthermore, our study also includes emissions from environmental disinfection, tap water usage, and pesticide use, broadening the sources and thus reducing the uncertainty of the inventory.

In our discussion within the main text in Section 4.1, we have addressed these comparisons with other literature.

[Comment]: 12. "Only this study and Yin et al. (2022) considered the emissions from cooking. The emission from cooking in this study is lower due to lower flue gas flux and shorter cooking durations". Qiu also considers cooking (Qiu, et al., Atmos. Chem. Phys. 2019, 19, 6737−6747).

Response: Thanks for your comment. Yin et al. (2022), Qiu et al. (2019), and this study all considered emissions from cooking. In this study, due to reference to national standards and actual conditions, lower smoke flow and shorter cooking time were used, and the proportion of $Cl^-$ in $PM_{2.5}$ obtained through local measurements was lower than

that of Qiu et al. (2019), thereby reducing cooking emissions. We have revised the text in Section 4.1 as follows: "Yin et al. (2022), Qiu et al. (2019b) and this study all considered emissions from cooking.In this study, we employed lower flue gas flow rates and shorter cooking durations following national standards and actual conditions, and lower proportions of Cl$^-$ in PM$_{2.5}$ based on the local measured data from the literature, which reduced the cooking emission compared with the other two studies."

Specific comments:

[Comment]: 13. Lines 35-40: The term "chlorine atoms" is defined as "Cl," which could lead to confusion with the "Cl free radical" mentioned in lines 45-50. To avoid any ambiguity, please consider clarifying this terminology or employing an alternate term for either "Cl free radical" or "chlorine atoms."

Response: Thank you very much for your suggestion. The entire text has been uniformly revised from "chlorine atoms" to "chlorine radials".

[Comment]: 14. Lines 53-54: The authors mentioned that "However, research on anthropogenic chlorine emission inventories in China is currently limited, and the temporal and spatial distribution of these emissions remains unclear". These descriptions are inaccurate. There are papers reporting temporal and spatial distribution of chlorine emissions (Fu et al., 2018; Yin et al., 2022).

Response: Thanks for pointing out this issue. We have removed this inaccurate description in the revised manuscript.

[Comment]: 15. Lines 54-57: The authors mentioned that "Consequently, anthropogenic chlorine emissions are rarely considered in numerical simulations of air quality, making it challenging to study the chemical mechanism of chlorine and quantify the contribution of anthropogenic chlorine emissions to ozone and other

pollutants using models". These descriptions are inaccurate. There are some papers that have already conducted the modeling study, but they are not properly cited (eg. Choi et al., Environ. Sci. Technol. 2020, 54, 13409−13418; Li et al., Environ. Sci. Technol. 2021, 55, 13625−13637; Wang et al., Atmos. Chem. Phys., 21, 13973–13996, 2021; Li et al., Journal of Geophysical Research: Atmospheres, 125, e2019JD032058. https://doi.org/10.1029/2019JD032058; Li et al., Journal of Geophysical Research: Atmospheres, 126, e2020JD034175. https://doi.org/10.1029/2020JD034175; Wang et al., Cite This: Environ. Sci. Technol. 2020, 54, 9908−9916; ⋯).

Response: Thanks for pointing out this issue. We have removed this inaccurate statement in Section 1. We have added the citations of these paper in Section 1 as follow: "some modeling studies (Choi et al., 2020; Li et al., 2021) have used the anthropogenic chlorine emission as inputs and found that the simulated concentrations of chlorine species (HCl and pCl) were underestimated against the observation, which suggests that there are large uncertainties or missing sources for the current emission estimation."

Reference:

Choi, M., Qiu, X., Zhang, J., Wang, S., Li, X., Sun, Y., Chen, J., Ying, Q.: Study of Secondary Organic Aerosol Formation from Chlorine Radical-Initiated Oxidation of Volatile Organic Compounds in a Polluted Atmosphere Using a 3D Chemical Transport Model, Environ Sci Technol., 54, 13409–13418, 2020.

Li, J., Zhang, N., Wang, P., Choi, M., Ying, Q., Guo, S., Lu, K., Qiu, X., Wang, S., Hu, M., Zhang, Y., Hu, J.: Impacts of chlorine chemistry and anthropogenic emissions on secondary pollutants in the Yangtze river delta region, Environmental Pollution, 287, 117624, 2021.

[Comment]: 16. Lines 75-80, lack punctuation: "and waste incineration The study pointed out."

Response: Thanks for pointing out this typo. We have added "." after "waste incineration" for a punctuation here in Section 1.

[Comment]: 17. Line 77, missing "." after "waste incineration".

Response: Thanks for pointing out this typo. We have added "." after "waste incineration" here in Section 1.

[Comment]: 18. Line 80: It defines "particulate Cl$^-$" as "(pCl)," which has already been defined in line 45. To avoid redundancy, please refrain from repeating this definition.

Response: Thank you for your suggestion. To avoid redundancy, we have removed the definition of pCl here in Section 1.

[Comment]: 19. Line 85: "chlorinecontaining" should be "chlorine-containing".

Response: Thanks for your suggestion. The word "chlorinecontaining" has been changed to "chlorine-containing" in Section 1.

[Comment]: 20. Line 115: The provided emission ratios of 0.84 and 0.11 for disinfectant use sources HOCl and Cl$_2$, respectively, require an explanation regarding the origin of this data.

Response: Thanks for pointing out this issue. In the revised manuscript, we mentioned the value of release ratios for usages of chlorine-containing disinfectants and pesticides with references in Section 2.2.6 and 2.2.7, respectively. Here, we removed this introduction in Section 2.1.

[Comment]: 21. Line 120: To ensure clarity, it's essential to delineate the distinction between "activity data" and "emission data." Additionally, providing a precise definition for "emission data" would enhance understanding.

Response: Thank you very much for your suggestion. The term "emission data" has

been revised to "emission factor" for clarification in the first paragraph of Section 2.2. We have thoroughly reviewed the entire article and made the necessary corrections.

[Comment]: 22. Lines 175-180, the numerical values for $\eta d$ and $\eta s$ need references.

Response: Thank you very much for your suggestion. The reference (Liu et al., 2018) has been added in Section 2.2.3. Besides, we have carefully checked the manuscript and provided references for all the numbers used for emission calculations.

Reference:

Liu, Y. M., Fan, Q., Chen, X. Y., Zhao, J., Ling, Z. H., Hong, Y. Y., Li, W. B., Chen, X. L., Wang, M. J., and Wei, X. L.: Modeling the impact of chlorine emissions from coal combustion and prescribed waste incineration on tropospheric ozone formation in China, Atmos Chem Phys, 18, 2709-2724, 10.5194/acp-18-2709-2018, 2018.

[Comment]: 23. Line 195 mentions that the proportion of open burning of solid waste varies by location. Are there specific values listed in the appendix or a reference for this information?

Response: Thank you very much for your suggestion. F represents the proportion of open burning of solid waste, which means the untreated portion. It is calculated using 1-f. f represents the treated proportion of solid waste, which is derived from the China Urban and Rural Construction Statistical Yearbook 2019 (National Bureau of Statistics, 2019). We have added the text in Section 2.2.3 as follows: "F represents the proportion of open burning of solid waste, which means the untreated portion (1-f). f represents the treated proportion of solid waste, which is derived from the China Urban and Rural Construction Statistical Yearbook 2019 (National Bureau of Statistics, 2019a). The F value varied in different provinces due to the imbalance of economy, urbanization, and garbage disposal technology popularization."

Reference:

National Bureau of Statistics: China Urban-Rural Construction Statistical Yearbook (2019), China Statistics Press, Beijing, China, 2019.

[Comment]: 24. Lines 285-305: It appears that formulas (15) and (16) might be repetitive. Considering that the study encompasses both semi-standard and non-standard swimming pools, totaling 72%, could these categories be combined for the purpose of calculation? The study initially classifies swimming pools into public and private, and subsequently differentiates them as indoor and outdoor pools. Is it accurate to assume that the ratios of indoor and outdoor pools are evenly distributed between public and private ones? Additionally, does the value assigned to outdoor pool openings throughout the year seem excessively high?

Response: Thanks for your comment. Formulas 15 and 16 are two different calculations for the volumes of public and private swimming pools, which are not repetitive. We agree that the semi-standard and non-standard swimming pools can be combined for calculation purposes. To avoid confusion, we have revised the text in Section 2.2.6d as follows: "Swimming pools include public swimming pools and private swimming pools, and they have different volumes. The volume of public swimming pools was calculated as follows:

$$V_i = n_i \times \sum_j (a_j \times b_j \times h_j \times r_j) \tag{15}$$

where i and j represent different provinces and size types. Swimming pool size types include standard, semi-standard/non-standard swimming pools. We assume that the sizes of semi-standard and non-standard swimming pools are the same. n is the number of swimming pools, and the provincial data comes from the State Sports General Administration (https://www.sport.gov.cn/, last access: 1 January 2023). a, b, and h are the length, width, and depth of the swimming pool with different size types, as shown in Table S11. r represents the proportion of different size types of swimming pools, with standard swimming pools accounting for 28%, and semi-standard/non-standard swimming pools accounting for 72% (Zhang, 2015).

The volume of private swimming pools was calculated as follows:

$$V_i = n_i \times a \times b \times h \tag{16}$$

where i represent different provinces. n is the number of swimming pools, and the provincial data are estimated based on the ratio of residents' income to the number of swimming pools following the method proposed by Li et al. (2020). a, b, and h are the length, width, and depth of the private swimming pool, as shown in Table S11."

Due to the lack of relevant literature research or statistical data, this study assumes that the proportion of indoor and outdoor swimming pools is evenly distributed between public and private swimming pools. This assumption may induce uncertainties, and if the data is available in the future, we will improve it accordingly.

As for the opening hours of outdoor swimming pools, we made a typo during the writing process of the article. We have corrected the text in Section 2.2.6d as follows: "D is the number of opening days for the swimming pool. The indoor swimming pool in this study is open all year round, and the outdoor swimming pool is only open in summer."

[Comment]: 25. Line 317: Considering the total health expenditure and the corresponding ratio, it is estimated to be 11,898.0 L in 2018. How is it derived?

Response: Thank you for your question. As reported, in 2007, Taizhou hospitals used an average of 2329.2 L of chlorine-containing disinfectants (Sun et al., 2007). Since there was no data available for the usage of chlorine-containing disinfectants in 2018, we assumed that its change is proportional to the total health cost in recent years. The amount of disinfectant usage is estimated using the formula $U_{2018} = U_{2007} \times C_{2018}/C_{2007}$, where U represents the amount of disinfectant usage, and C represents the total health cost. The total health cost can be obtained from the China Health Statistical Yearbook (National Bureau of Statistics, 2008, 2019). As a result, the estimated usage of disinfectants in 2018 is 11898.0 L. It's worth noting that the inflation and price fluctuations may induce the uncertainty of this estimation method. We have added the text in Section 2.2.6e as follows: "Due to the absence of data in 2018, we assumed that its change is proportional to the total health cost in recent years. The amount of

disinfectant usage is estimated using the formula $U_{2018}=U_{2007} \times C_{2018}/C_{2007}$, where U represents the amount of disinfectant usage, and C represents the total health cost. The total health cost can be obtained from the China Health Statistical Yearbook (National Health Commission of the People's Republic of China, 2019, 2008). As a result, the usage of disinfectants in 2018 is estimated to be 11898.0 L".

Reference:

National Health Commission of the People's Republic of China: China Health Statistical Yearbook (2008), Peking Union Medical College Press, Beijing, China, 2008.

National Health Commission of the People's Republic of China: China Health Statistical Yearbook (2019), Peking Union Medical College Press, Beijing, China, 2019.

[Comment]: 26. Line 325, formula (19) does not provide the chlorine disinfectant concentration for the aquaculture industry. What are the proportions of HOCl and $Cl_2$ in this case?

Response: Thanks for pointing out this issue. The release ratios of HOCl and $Cl_2$ are provided in this formula. They are 0.84 and 0.11 according to the study of Wong et al. (2017). We have added the text in Section 2.2.6e as follows: "R is the release ratio of HOCl (0.84) and $Cl_2$ (0.11) (Wong et al., 2017)".

Reference:

Wong, J. P. S., Carslaw, N., Zhao, R., Zhou, S., and Abbatt, J. P. D.: Observations and impacts of bleach washing on indoor chlorine chemistry, Indoor Air, 27, 1082-1090, 2017.

[Comment]: 27. Line 340: It's possible that chlorine disinfectant use in household toilets is lower compared to public restrooms. Could the assumption of a 2 times higher chlorine disinfectant use in public toilets potentially be an overestimation? I

recommend estimating the quantity of chlorine disinfectant utilized per household (e.g., per bottle of bleach) based on population, and subsequently comparing it against the emission estimate rooted in the 2 times higher value. This analysis can help identify any potential instances of overestimation.

Response: Thanks for your suggestion. Due to the absence of data pertaining to disinfectant usage at the household level, coupled with the inherent variability in disinfectant consumption among households, the results of our study are subject to considerable uncertainty. In light of these limitations, we have opted to adopt the approach outlined by Li et al. (2020), as a temporary solution to address this challenge.

[Comment]: 28. Line 375, please provide the emission ratios for HOCl and $Cl_2$ during pesticide application.

Response: Thank you for your question. The emission ratios for HOCl and $Cl_2$ are 0.84 and 0.11, respectively, during pesticide application. These values were adopted according to the study of Wong et al. (2017) and Yi et al. (2021). We have added this information in Section 2.2.7 as follows: "R is the release ratio of HOCl (0.84) and $Cl_2$ (0.11) during the pesticide application (Wong et al., 2017; Yi et al., 2021)".

Reference:

Wong, J. P. S., Carslaw, N., Zhao, R., Zhou, S., and Abbatt, J. P. D.: Observations and impacts of bleach washing on indoor chlorine chemistry, Indoor Air, 27, 1082-1090, 2017.

Yi, X., Yin, S., Huang, L., Li, H., Wang, Y., Wang, Q., Chan, A., Traoré, D., Ooi, M. C. G., Chen, Y., Allen, D. T., and Li, L.: Anthropogenic emissions of atomic chlorine precursors in the Yangtze River Delta region, China, Sci. Total Environ., 771, 144644, https://doi.org/10.1016/j.scitotenv.2020.144644, 2021.

[Comment]: 29. Lines 442-448, the authors presented per-unit-area and per-capita emissions. However, it remains unclear what reasons contribute to these results? Within

the discussion section, the paper predominantly showcases data results, yet falls short in delving into a comprehensive exploration of the underlying reasons.

Response: Thank you for your suggestion. We have provided the reasons contributing to the provincial variations of per-unit-area and per-capita emissions in the revised manuscript. We have added the discussion in Section 3.3 as follows: "For the per-unit-area emission intensity, Shandong is the province with the highest emission intensity of HCl (238.13 kg km$^{-2}$) and fine particulate Cl$^-$ (132.05 kg km$^{-2}$), which is attributed to its relatively higher emission but smaller area. Shanghai has the highest emission intensity of Cl$_2$ (60.07 kg km$^{-2}$) and HOCl (419.48 kg km$^{-2}$), which is due to its small area. For the per-capita emission intensity, Heilongjiang has the highest emission intensity of HCl (1014.21g per people) and fine particulate Cl$^-$ (720.31 g per people) due to its highest emission across the country. Ningxia is the province with the highest emission intensity of Cl$_2$ (39.01 g per people) due to its low population. Shanghai is the province with the highest emission intensity HOCl (109.72 g per people) due to its relatively higher emission but lower population".

[Comment]: 30. Line 473-475, it is stated that "The emissions of HCl and fine particulate Cl exhibit relatively higher levels during early summer and autumn, coinciding with the frequent occurrence of biomass burning". However, it's important to note that the time period of biomass burning varies across different regions.

Response: Thanks for your comment, which we agree. In the revised manuscript, we used the fire location and its fire radiation power data from the Himawari-8 satellite data to allocate the temporal distribution of HCl and pCl emissions from biomass open burning in various provinces. We have added this description in Section 2.4 as follows: "Based on the fire location and its fire radiation power over the cropland from the Himawari-8 satellite data, we performed temporal allocation of chlorine emissions from biomass burning for each province". We have created new temporal distribution maps (Figure 6) and provided the temporal distribution maps for the seven major regions in mainland China in the supplementary information (Figures S5 and S6). The results

show that the variation of open biomass burning emissions presents regional differences. We have added the discussion in Section 3.5 as follows: "Figure 6 shows the temporal variation of anthropogenic emissions for different chlorine species. For HCl and pCl, the emission in mainland China presents a bimodal variation. A remarkable peak is in early spring (February to April), and a small peak is in early autumn (August to October). The high emission in these months is attributed to the biomass burning emission with active agricultural activities. In contrast, emissions from other sectors remain relatively stable throughout the year. It's worth noting that the monthly variations vary across different regions because of the varied period of biomass burning, as shown in Fig. S5 and S6. For example, in Northeast China (Liaoning, Jilin, and Heilongjiang), where extensive straw burning occurs before crop planting, emissions are elevated in spring only".

[Comment]: 31. Line 544-545, "The inventory can be enhanced by including emissions from other anthropogenic activities that release chlorine. For example, the disposal and combustion of medical waste, which often contains high levels of plastic, can result in the release of significant amounts of active chlorine" isn't medical waste included in this study?

Response: Thank you for your question. This inventory currently includes the emissions from domestic waste treatment, in which medical waste is excluded. We have added the text in Section 2.2.3 for clarification as follows: "Currently, only the emissions from domestic waste are considered in this study". We acknowledge the value of your suggestion and plan to incorporate it into the inventory in the future for comprehensiveness.

[Comment]: 32. Line 554-555: "In this study, we developed a Chinese anthropogenic chlorine emissions inventory (ACEIC 2018) using emission factors mainly based on local measurements", this is inaccurate, as there are no measurements presented in this

paper.

Response: Thanks for pointing out this issue. The emission factors for emission calculation were mainly based on the local measured and survey data from the literature. We have clarified the text in Section 5 as follows: "In this study, we developed a Chinese anthropogenic chlorine emissions inventory (ACEIC 2018) using emission factors mainly based on local measured and survey data from the literature."

[Comment]: 33. In section 3.3, it is suggested to provide reasons for higher chlorine emissions in different provinces to enhance the results analysis.

Response: Thank you for your suggestion, which we accept. We have revised the text in Section 3.3 as follows: "Regarding HCl emissions, Heilongjiang (38.27 Gg), Shandong (38.10 Gg), Henan (36.05 Gg), Hebei (32.46 Gg), and Hunan (24.45 Gg) emerge as the top five contributing provinces. They account for 8.4%, 8.4%, 7.9%, 7.2%, and 5.4% of the total emissions, respectively. The elevated emissions in Heilongjiang, Shandong, and Henan are attributed to the major contributions of biomass burning with higher agricultural production, which accounts for 77%, 54%, and 58%, respectively. For Hebei and Hunan, the higher emissions from industrial production are the major contributors, accounting for 40% and 31%, respectively. The top five contributors to fine particulate $Cl^-$ emissions are Heilongjiang (27.18 Gg), Henan (21.60 Gg), Shandong (21.13 Gg), Hebei (15.46 Gg), and Anhui (14.67 Gg). In these provinces, higher biomass burning emissions induced by active agricultural activities dominate the total emission. $Cl_2$ emissions are predominantly attributed to Guangdong (1.40 Gg), Shandong (1.22 Gg), Hebei (1.09 Gg), Jiangsu (1.00 Gg), and Hunan (0.91 Gg). The top five provinces contributing to HOCl emissions are Guangdong (8.61 Gg), Jiangsu (5.50 Gg), Shandong (4.86 Gg), Zhejiang (4.04 Gg), and Sichuan (3.39 Gg). Due to the large population and developed economy that stimulates the need for disinfection processes, provinces such as Guangdong, Shandong, and Jiangsu have relatively high emissions of $Cl_2$ and HOCl".

[Comment]: 34. Line 740 mentions missing information on the meaning of the green line in Figure 4.

Response: Thanks for pointing out this issue. The green line in Figure 4 represents the emissions for each province. We have added this information in the caption of Figure 4 as follows: "Figure 4 Emissions (green line) and contribution proportions of HCl (a), fine particulate Cl$^-$ (b), Cl$_2$ (c), and HOCl (d) by province in 2018."

[Comment]: 35. Line 760 suggests changing "Power" in the "Subsector" in Table 3 to "coal combustion."

Response: Thank you for your suggestion. We have replaced "Power" under the "Subsector" category with "Coal combustion" in Table 3.

[Comment]: 36. Upon observing Figure S1, it raises the question of why the Per-unit-area emissions of Cl$_2$ and HOCl are notably elevated in Shanghai.

Response: Thank you for your question. The notably elevated per-unit-area emission intensity of Cl$_2$ and HOCl in Shanghai is due to its small area. We have added this discussion in Section 3.3 as follows: "Shanghai has the highest emission intensity of Cl$_2$ (60.07 kg km$^{-2}$) and HOCl (419.48 kg km$^{-2}$), which is due to its small area".

---

## Author Response (AR2)

The most important of this study is to provide more accurate chlorine emission inventory (with significantly reduced uncertainties) than previous studies. The authors also pointed out that some modeling studies have used the anthropogenic chlorine emission as inputs and found that the simulated concentrations of chlorine species (HCl and pCl) were underestimated against the observations, suggesting that there are large uncertainties or missing sources for the current emission estimation. Therefore, the chlorine emission inventory shall be proved to be more reasonable and has been cross-checked. Otherwise, any study is just a big homework to duplicate or compile previous works. Unfortunately, I do see this study is lack of solid basis or significant improvement in terms of more accurate emissions factors, etc. This flaw makes this study less meaningful.

When comparing with other works, the authors frequently attributed the difference to the use of different methods / different emission factors, without further in-depth explanations. The authors emphasize frequently that this updated inventory considered more anthropogenic sources, used more localized emission factors, and adopted more refined estimation methods. However, which one is better? Can the authors prove that the current emission factors / methods are more accurate? Can the authors prove that the uncertainties have been significantly reduced? For example, comparing results with observations?

Response: Thank you for your feedback. We have improved the emission inventory in the following aspects: (1) The anthropogenic chlorine emission inventory has been updated from the year 2018 to 2019 in this study. (2) Added emissions from medical waste disposal, an aspect often overlooked in other studies. With increasing public awareness of hygiene, emissions from medical waste disposal have become more important. (3) For biomass burning, adjustments were made to both household burning rates and open burning rates. Specifically, for biomass open burning ratios, this study re-calibrated the ratios from the research by Zhou et al. (2017) based on changes in fire

radiative power over croplands from MODIS satellite fire point data. Additionally, through literature research and based on Liu et al. (2022), we used statistical linear trend analysis to estimate household burning ratios for 2019. (4) Regarding car washing water, as opposed to Li et al. (2020), which assumed uniform water usage for each car wash, this study considered different types of vehicles with varying water usage and washing frequencies to provide a more realistic estimation.

With these improvements, the uncertainty of the emission inventory is significantly reduced compared with previous research. Additionally, we validated the emission inventory by incorporating it into the WRF-CMAQ model for a one-year simulation and comparing the simulated chlorine species with the observations. This cross-validation further demonstrates the relatively faithful estimation of chlorine emission in this study. The validation can be found in section 4.1.

Major comments:

[Comment]: 1. In all the formulars, there are even no units for each parameter. In the calculation method of Cl2 and HOCl emissions, they do not consider the available chlorine parameters and Molecular weight and other indicators.

Response: Thank you for your comment. The data referenced in the formulas are mostly provided in the SI manuscript, and units are specified for these data. Furthermore, parameters with units in the main text have been appropriately modified and supplemented. In this study, the term "chlorine addition" refers to the effective chlorine content in water after the addition of disinfectants. Therefore, we believe there is no need for additional parameters such as effective chlorine and molecular weight.

[Comment]: 2. For Industrial Production Process, there are chemical industries producing disinfectants, which will also emit HOCl but neglected. For spatial allocation of industrial production, the paper assumed a uniform emission for them and spatially allocated the provincial emissions evenly to each point!

Response: Thank you for your comment. Unfortunately, we currently lack data on disinfectant production, so emissions from the production of disinfectants cannot be considered at the moment. Additionally, we do not have specific production scale data for each factory. If we obtain this information in the future, we will certainly update the inventory more accurately.

[Comment]: 3. Environmental disinfection: The quantity of disinfectant utilized in each hospital is a very important parameter to estimate $Cl_2$/HOCl emissions from hospitals. However, the paper just citied a very old data in Taizhou in 2007 and expanded to year 2018 by using the formula $U_{2018}=U_{2007} \times C_{2018}/C_{2007}$.

Response: Thank you for your inquiry. With enhanced awareness of hygiene in hospitals, both the usage of disinfectants and overall healthcare costs tend to increase. The overall healthcare costs can to some extent reflect the changes in disinfectant usage. Therefore, in the absence of specific studies on the quantity of disinfectant used in hospitals, this study's estimation of the disinfectant usage for the year 2019 based on changes in overall healthcare costs is considered reasonable.

[Comment]: 4. Biomass burning: The study used the proportion of open biomass burning of Zhou et al., (2017) to estimate emissions of HCl and pCl from biomass burning sources. However, China has made great efforts to ban open straw burning, will this method introduce overestimations?

Response: Thank you for your suggestion. Consequently, regarding the biomass open burning ratio, this study re-adjusted the ratios proposed by Zhou et al. (2017) based on the variation in fire radiative power over croplands from MODIS satellite fire point data (https://modis.gsfc.nasa.gov). In addition, through a literature review and drawing on the research by Liu et al. (2022), we estimated the household burning ratio for the year 2019 using a statistical linear trend. Specific numerical values can be found in Table S8.

Liu, Y., Zhao, H., Zhao, G., Zhang, X., Xiu, A.: Carbonaceous gas and aerosol emissions from biomass burning in China from 2012 to 2021, Journal of Cleaner Production, 362, 132199, 2022.

[Comment]: 5. Usage of pesticide: Most data are citied from the year 2016.

Response: Thank you for your inquiry. Regarding the use of pesticides, the activity level data in this study is subject to variation. However, the emission factor, as the pesticide application process typically occurs in open settings, remains constant in its release rate. Therefore, we have set the emission factor for pesticides to be constant and not vary with the years.

[Comment]: 6. Chlorine-containing disinfectant: Table S5 gives the emission factors of $Cl_2$ and $HOCl$ from various disinfectant sources. Are they inconsistent with the calculations of formulas 3 and 12? If they know the emission factor data for each type of source, the emissions can be calculated directly with the activity level data.

Response: Thank you for your inquiry. The emission factors column in the original Table S5 was not obtained through literature research but was calculated and processed by us. In order to avoid any potential confusion for readers, we have removed that column of data. The table has been revised, and you can find the updated information in Table S11.

Anonymous Referee #2

Review comments on the manuscript titled "ACEIC: a comprehensive anthropogenic chlorine emission inventory for China"

The authors have addressed the points raised by reviewers and improved the manuscript. I only have the minor comments below for techincal corrections:

[Comment]: Line 143, the term of "chlorine content" is confusing. Maybe use something like mass ratio of chlorine in chlorine species.

Response: Thank you for your suggestion. The phrase "chlorine content in chlorine species" has been revised to "mass ratio of chlorine in chlorine species."

[Comment]: In Section 4.1, please cite the references properly when you showed values from literature.

Response: Thank you for your suggestion. The citations have been added as suggested.

[Comment]: Fig 5, I still think a unit of Mg/grid is meaningless. You can easily transfer the unit to Mg/km^2.

Response: Thank you for your suggestion. The units for Figure 5 and Figures S3, S4 have been changed to kg km$^{-2}$ yr$^{-1}$.

---

## Author Response (AR3)

**Response to the comment from Anonymous referee #4**

This study aims to provide a comprehensive inventory of chlorine emissions by updating the previous inventory to include data from a more recent year and expanding the range of species considered, as well as the number of anthropogenic sources. This paper provides valuable data on chlorine emissions in China. Despite the considerable effort involved, I suggest that the authors enhance the foundational support for the new additions and provide more evidence to demonstrate significant advancements in emission inventory calculation methods. Strengthening these aspects will greatly improve the reliability and impact of the study. My main suggestions for improving innovation and reducing uncertainty are as follows:

[Comment 1]: A significant portion of the chlorine emission sources in the study are indoors. However, the exchange of gases between indoor and outdoor environments is influenced by various factors, such as building structures, climate conditions, and building usage across different regions of China. It is recommended that the authors fully consider the impact of these factors.

Response: Thank you for your valuable suggestion. Many previous studies demonstrated that the indoor air pollutants can significantly affect outdoor air quality through indoor-outdoor exchange, but quantifying this impact is challenging and remains uncertainties (Santiago et al., 2022; Alsamrai et al., 2024). These uncertainties arise from various factors, including the complexity of emission sources, variability in building ventilation systems, and differences in geographic and climatic conditions. Here, to simplify, we assumed that all the indoor chlorine gases (e.g. environmental disinfection, indoor swimming pool) released into the atmosphere due to the rapid air exchange between indoor and outdoor during the ventilation process (Huang, 2012; Tang, 2003). Although this treatment may overestimate their impact on ambient concentrations, our modeling results show general agreement with the observation. We have discussed this limitation in the manuscript.

Revision in the updated manuscript:

1) Line 671-676: "(2) Complexity of indoor-outdoor air exchange: The study simplifies the treatment of some indoor chlorine emissions (e.g., environmental disinfection) by assuming rapid air exchange between indoor and outdoor environments. However, the actual exchange rate can vary significantly due to differences in building structures, ventilation systems, and climatic conditions. This variability can lead to uncertainties in estimating the impact of indoor emissions on outdoor air quality. Future research should incorporate more detailed assessments of these factors to improve the accuracy of emission estimates. "

Reference:

Alsamrai, O., Redel-Macias, M. D., Pinzi, S., and Dorado, M. P.: A Systematic Review for Indoor and Outdoor Air Pollution Monitoring Systems Based on Internet of Things, Sustainability, 16, 4353, 2024.

Huang, Y.: Study of the Natural Ventilation Strategy of Hospital Clinic Waiting in Lingnan Regions, M.S. thesis, South China University of Technology, China, 125 pp., 2012.

Tang, J.: Design of the Air-Conditioner for Chamber Indoor Swimming Pool, Mechanical and Electrical Equipment, 5, 17-20, 2003 (in Chinese).

Santiago, J. L., Rivas, E., Buccolieri, R., Martilli, A., Vivanco, M. G., Borge, R., Carlo, O. S., and Martín, F.: Indoor-outdoor pollutant concentration modelling: a comprehensive urban air quality and exposure assessment, Air Qual Atmos Health, 15, 1583-1608, 10.1007/s11869-022-01204-0, 2022.

[Comment 2]: Adding new emission sources or further subdividing the emission sources does not necessarily lead to better results. Inaccurate descriptions of the spatial and temporal distribution of sources can increase uncertainty. Given chlorine's high atmospheric reactivity, spatial and temporal distribution is crucial. For example, point

sources and area sources of chlorine emissions have significantly different atmospheric chemical impacts. The study's simplistic allocation of some point source emissions to entire areas may be inappropriate; additionally, the temporal distribution of pesticide use is influenced by crop types, climate, and geographical conditions, which the study does not detail.

Response: Thanks for your valuable suggestion. Firstly, regarding the spatial allocation of some point sources, such as residential coal combustion and biomass household burning, we treated them as areas sources and used the total/urban/rural population as a proxy for disaggregating emissions. Although this treatment may induce uncertainties, it is commonly adopted for those point sources without available specific location information in the development of national emission inventories in previous studies. As summarized in Table S4 by Li et al. (2017), for the well-known emission inventories of MEIC (http://meicmodel.org.cn), REAS2 (Kurokawa et al., 2013), PKU-NH$_3$ (Huang et al., 2012), ANL-India (Lu et al., 2011), CAPSS (Lee et al., 2011), total/urban/rural population served as a proxy for spatial allocations of some point sources (residential, industrial, solvent use, agriculture, etc.).

Secondly, regarding the temporal distribution of pesticides, we used the monthly pesticide production as the temporal allocation factors. The temporal variation of pesticide production can somewhat reflect the monthly emission characteristics of pesticide usage. We have provided this detailed information in Section 2.4.

Overall, there is a long way to go to reduce the uncertainties of air pollutant emission inventories currently. Further analysis and research are needed to reduce these uncertainties if specific sources information are available and novel estimation methods are proposed in the future. We have discussed the limitation of this inventory in the manuscript.

Revision in the updated manuscript:
1) Line 676-682: "(3) Spatial and temporal distribution of emissions: The study employs generalized proxies, such as population distribution, for spatial allocation of emissions from point sources like residential coal combustion and biomass burning.

This approach may not accurately capture the localized nature of emissions and their impacts. Additionally, the temporal allocation of pesticide emissions based on monthly production does not fully account for regional variations in crop types and climate. More precise data on spatial and temporal allocation factors, such as detailed information on the operating scales of point sources, are needed to reduce uncertainties and enhance the accuracy of the inventory. "

Reference:

Huang, X., Song, Y., Li, M., Li, J., Huo, Q., Cai, X., Zhu, T., Hu, M., and Zhang, H.: A high‐resolution ammonia emission inventory in China, Global Biogeochem Cy, 26, 2012.

Kurokawa, J., Ohara, T., Morikawa, T., Hanayama, S., Janssens-Maenhout, G., Fukui, T., Kawashima, K., and Akimoto, H.: Emissions of air pollutants and greenhouse gases over Asian regions during 2000–2008: Regional Emission inventory in ASia (REAS) version 2, Atmos Chem Phys, 13, 11019-11058, 2013.

Lee, D., Lee, Y.-M., Jang, K.-W., Yoo, C., Kang, K.-H., Lee, J.-H., Jung, S.-W., Park, J.-M., Lee, S.-B., and Han, J.-S.: Korean national emissions inventory system and 2007 air pollutant emissions, Asian Journal of atmospheric environment, 5, 278-291, 2011.

Li, M., Zhang, Q., Kurokawa, J., Woo, J. H., He, K. B., Lu, Z. F., Ohara, T., Song, Y., Streets, D. G., Carmichael, G. R., Cheng, Y. F., Hong, C. P., Huo, H., Jiang, X. J., Kang, S. C., Liu, F., Su, H., and Zheng, B.: MIX: a mosaic Asian anthropogenic emission inventory under the international collaboration framework of the MICS-Asia and HTAP, Atmos Chem Phys, 17, 935-963, 10.5194/acp-17-935-2017, 2017.

Lu, Z., Zhang, Q., and Streets, D. G.: Sulfur dioxide and primary carbonaceous aerosol emissions in China and India, 1996–2010, Atmos Chem Phys, 11, 9839-9864, 2011.

[Comment 3]: The authors should provide stronger evidence to show that this more

detailed method significantly reduces uncertainty. Some of these more detailed activity amounts and emission factors have limited sources. For example, the quantity of disinfectant utilized, which is cited from only one hospital, may not be sufficiently convincing.

Response: We appreciate your valuable suggestion. Firstly, the quantity of disinfectant utilized in a hospital to estimate emissions in this study is based on the survey results of Sun et al. (2007), with adjustments for annual variations. It is reported that the amount of disinfectant used in five hospitals (not only one hospital) in Taizhou ranged from 1997 to 2704 L, with an average of 2329.2L. The small deviation among these five hospitals shows that this average value is generally representative for the emission estimation in this study. Nevertheless, specific dataset about the quantify of disinfectant utilized in each hospital in China is warranted to estimate chlorine emissions accurately, but corresponding survey data are scarce currently. We have discussed this limitation in the manuscript.

Revision in the updated manuscript:

1) Line 666-671: "The potential limitations of the ACEIC emission inventory that require further refinement are summarized as follows. (1) Limited sources for some activity data and emission factors: For example, for estimating chlorine emissions from hospital disinfection, the average disinfectant usage derived from five hospitals may not accurately reflect the variability in usage across different regions and facility types. There is a need for further localized investigation on activity data and experimental studies on emission factors. This is particularly important for sectors such as waste treatment and usage of chlorine-containing disinfectant, where more localized observation data is needed to improve the accuracy of emission estimates. "

Reference:

Sun, Y., Tian, F., Sun, Z., Jiang, W., Wu, B., Xu, Z., and Gu, J.: Investigation and Analysis of the Current Condition of Applying Disinfectants in City Hospitals of Taizhou, Modern Preventive Medicine (in Chinese), 4738-4741, 2007.

Overall, my view aligns with previous reviewers that the methodologies and details of this study could benefit from further refinement. I encourage the authors to pursue additional investigation and experimental support.

Response: Thanks for your valuable comment. We recognize that this chlorine emission inventory still has some limitations which may induce uncertainties. In the future, we will continue to refine and update this inventory with further investigation and experimental support.

---

## Author Response (AR4)

Dear Editor,

Thanks for your recognization of this paper. Some typos have been corrected in the manuscript and supplementary information.

1) Line 295 in manuscript: "obtainted" to "obtained".
2) Table S12 in SI: "sha3nxi" to "shaanxi".

Best regards,
Yiming Liu